# Spectral Norm Regularization of Orthonormal Representations for Graph Transduction

**Rakesh Shivanna**
Google Inc.
Mountain View, CA, USA
rakeshshivanna@google.com

**Bibaswan Chatterjee**
Dept. of Computer Science & Automation
Indian Institute of Science, Bangalore
bibaswan.chatterjee@csa.iisc.ernet.in

**Raman Sankaran, Chiranjib Bhattacharyya**
Dept. of Computer Science & Automation
Indian Institute of Science, Bangalore
ramans,chiru@csa.iisc.ernet.in

**Francis Bach**
INRIA - Sierra Project-team
École Normale Supérieure, Paris, France
francis.bach@ens.fr

## Abstract

Recent literature [1] suggests that embedding a graph on an unit sphere leads to better generalization for graph transduction. However, the choice of optimal embedding and an efficient algorithm to compute the same remains open. In this paper, we show that orthonormal representations, a class of unit-sphere graph embeddings are PAC learnable. Existing PAC-based analysis do not apply as the VC dimension of the function class is infinite. We propose an alternative PAC-based bound, which do not depend on the VC dimension of the underlying function class, but is related to the famous Lovász $\vartheta$ function. The main contribution of the paper is SPORE, a SPectral regularized ORthonormal Embedding for graph transduction, derived from the PAC bound. SPORE is posed as a non-smooth convex function over an *elliptope*. These problems are usually solved as semi-definite programs (SDPs) with time complexity $O(n^6)$. We present, Infeasible Inexact proximal (IIP): an Inexact proximal method which performs subgradient procedure on an approximate projection, not necessarily feasible. IIP is more scalable than SDP, has an $O(\frac{1}{\sqrt{T}})$ convergence, and is generally applicable whenever a suitable approximate projection is available. We use IIP to compute SPORE where the approximate projection step is computed by FISTA, an accelerated gradient descent procedure. We show that the method has a convergence rate of $O(\frac{1}{\sqrt{T}})$.

The proposed algorithm easily scales to 1000's of vertices, while the standard SDP computation does not scale beyond few hundred vertices. Furthermore, the analysis presented here easily extends to the multiple graph setting.

## 1 Introduction

Learning problems on graph-structured data have received significant attention in recent years [11, 17, 20]. We study an instance of graph transduction, the problem of learning labels on vertices of simple graphs[1]. A typical example is webpage classification [20], where a very small part of the entire web is manually classified. Even for simple graphs, predicting binary labels of the unlabeled vertices is NP-complete [6].

More formally: let $G = (V, E)$, $V = [n]$ be a simple graph with unknown labels $\mathbf{y} \in \{\pm 1\}^n$. Without loss of generality, let the labels of first $m \in [n]$ vertices be observable, let $u := n - m$.

Let $\mathbf{y}_S$ and $\mathbf{y}_{\bar{S}}$ be the labels of $S = [m]$ and $\bar{S} = V \backslash S$. Given $G$ and $\mathbf{y}_S$, the goal is to learn soft predictions $\hat{\mathbf{y}} \in \mathbb{R}^n$, such that $er_{\bar{S}}^{\ell}[\hat{\mathbf{y}}] := \frac{1}{|\bar{S}|} \sum_{j \in \bar{S}} \ell(y_j, \hat{y}_j)$ is small, where $\ell$ is any loss function. The following formulation has been extensively used [19, 20]

$$\min_{\hat{\mathbf{y}} \in \mathbb{R}^n} \; er_S^{\ell}[\hat{\mathbf{y}}] + \lambda \hat{\mathbf{y}}^{\top} \mathbf{K}^{-1} \hat{\mathbf{y}}, \tag{1}$$

where $\mathbf{K}$ is a graph-dependent kernel and $\lambda > 0$ is a regularizer constant. Let $\hat{\mathbf{y}}^*$ be the solution to (1), given $G$ and $S \subseteq V$, $|S| = m$. [1] proposed the following generalization bound

$$\mathbb{E}_{S \subseteq V}\left[er_{\bar{S}}^{\ell}[\hat{\mathbf{y}}^*]\right] \leq c_1 \inf_{\hat{\mathbf{y}} \in \mathbb{R}^n} \left[er_V^{\ell}[\hat{\mathbf{y}}] + \lambda \hat{\mathbf{y}}^{\top} \mathbf{K}^{-1} \hat{\mathbf{y}}\right] + c_2 \left(\frac{tr_p(\mathbf{K})}{\lambda |S|}\right)^p, \tag{2}$$

where $c_1, c_2$ are dependent on $\ell$ and $tr_p(\mathbf{K}) = \left(\frac{1}{n} \sum_{i \in [n]} \mathbf{K}_{ii}^p\right)^{1/p}$, $p > 0$. [1] argued that $tr_p(\mathbf{K})$ should be a constant and can be enforced by normalizing the diagonal entries of $\mathbf{K}$ to be 1. This is an important advice in graph transduction, however it is to be noted that the set of normalized kernels is quite large and (2) gives little insight in choosing the optimal kernel.

Normalizing the diagonal entries of $\mathbf{K}$ can be viewed geometrically as embedding the graph on a unit sphere. Recently, [16] studied a rich class of unit sphere graph embeddings, called orthonormal representations [13], and find that it is statistically consistent for graph transduction. However, the choice of the optimal orthonormal embedding is not clear. We study orthonormal representations for the following equivalent [19] kernel learning formulation of (1), with $C = \frac{1}{\lambda m}$,

$$\omega_C(\mathbf{K}, \mathbf{y}_S) = \max_{\alpha \in \mathbb{R}^n} \sum_{i \in S} \alpha_i - \frac{1}{2} \sum_{i,j \in S} \alpha_i \alpha_j y_i y_j K_{ij} \;\; \text{s.t.} \;\; 0 \leq \alpha_i \leq C \, \forall i \in S, \; \alpha_j = 0 \, \forall j \notin S, \tag{3}$$

from a probably approximately correctly (PAC) learning point of view. Note that the final predictions are given by $\hat{y}_i = \sum_{j \in S} K_{ij} \alpha_j^* y_j \; \forall i \in [n]$, where $\alpha^*$ is the optimal solution to (3).

**Contributions.** We make the following contributions:
- Using (3) we show the class of orthonormal representations are efficiently PAC learnable over a large class of graph families, including power-law and random graphs.
- The above analysis suggests that spectral norm regularization could be beneficial in computing the best embedding. To this end we pose the problem of SPectral norm regularized ORthonormal Embedding (**SPORE**) for graph Transduction, namely that of minimizing a convex function over an *elliptope*. One could solve such problems as SDPs which unfortunately do not scale well beyond few hundred vertices.
- We propose an infeasible inexact proximal (IIP) method, a novel projected subgradient descent algorithm, in which the projection is approximated by an inexact proximal method. We suggest a novel approximation criteria which approximates the proximal operator for the support function of the feasible set within a given precision. One could compute an approximation to the projection from the *inexact proximal point* which may not be feasible hence the name IIP. We prove that IIP converges to the optimal minimum of a non-smooth convex function with rate $O(1/\sqrt{T})$ in $T$ iterations.
- The IIP algorithm is then applied to the case when the set of interest is composed of the intersection of two convex sets. The proximal operator for the support function of the set of interest can be obtained using the FISTA algorithm, once we know the proximal operator for the support functions of the individual sets involved.
- Our analysis paves the way for learning labels on multiple graphs by using the embedding by adopting an MKL style approach. We present both algorithmic and generalization results.

**Notations.** Let $\| \cdot \|$, $\| \cdot \|_F$ denote the Euclidean and Frobenius norm respectively. Let $\mathcal{S}_n$ and $\mathcal{S}_n^+$ denote the set of $n \times n$ square symmetric and square symmetric positive semi-definite matrices respectively. Let $\mathbb{R}_+^n$ be a non-negative orthant. Let $\mathcal{S}^{n-1} = \left\{ \mathbf{u} \in \mathbb{R}_+^n \; \big| \; \|\mathbf{u}\|_1 = 1 \right\}$ denote the $n-1$ dimensional simplex. Let $[n] := \{1, \ldots, n\}$. For any $\mathbf{M} \in \mathcal{S}_n$, let $\lambda_1(\mathbf{M}) \geq \ldots \geq \lambda_n(\mathbf{M})$ denote its Eigenvalues. We denote the adjacency matrix of a graph $G$ by $\mathbf{A}$. Let $\bar{G}$ denote the complement graph of $G$, with the adjacency matrix $\bar{\mathbf{A}} = \mathbf{1}\mathbf{1}^{\top} - \mathbf{I} - \mathbf{A}$; where $\mathbf{1}$ is a vector of all 1's, and $\mathbf{I}$ is the identity matrix. Let $\mathcal{Y} = \{\pm 1\}, \hat{\mathcal{Y}} = \mathbb{R}$ be the label and soft-prediction spaces over $V$. Given $y \in \mathcal{Y}$

and $\hat{y} \in \widehat{\mathcal{Y}}$, we use $\ell^{0\text{-}1}(y,\hat{y}) = \mathbb{1}[y\hat{y} < 0], \ell^{hng}(y,\hat{y}) = (1 - y\hat{y})_+^2$ to denote 0-1 and hinge loss respectively. The notations $O, o, \Omega, \Theta$ will denote standard measures in asymptotic analysis [4].

**Related work.** [1]'s analysis was restricted to Laplacian matrices, and does not give insights in choosing the optimal unit sphere embedding. [2] studied graph transduction using PAC model, however for graph orthonormal embeddings, there is no known sample complexity estimate. [16] showed that working with orthonormal embeddings leads to consistency. However, the choice of optimal embedding and an efficient algorithm to compute the same remains an open issue. Furthermore, we show that [16]'s sample complexity estimate is sub-optimal.

**Preliminaries.** An *orthonormal embedding* [13] of a simple graph $G = (V,E)$, $V = [n]$, is defined by a matrix $\mathbf{U} = [\mathbf{u}_1, \ldots, \mathbf{u}_n] \in \mathbb{R}^{d \times n}$ such that $\mathbf{u}_i^\top \mathbf{u}_j = 0$ whenever $(i,j) \notin E$ and $\|\mathbf{u}_i\| = 1 \; \forall i \in [n]$. Let $Lab(G)$ denote the set of all possible orthonormal embeddings of the graph $G$, $Lab(G) := \{\mathbf{U} \mid \mathbf{U}$ is an orthonormal embedding$\}$. Recently, [8] showed an interesting connection to the set of graph kernel matrices

$$\mathcal{K}(\mathbf{G}) := \{\mathbf{K} \in \mathcal{S}_n^+ \mid K_{ii} = 1, \forall i \in [n]; K_{ij} = 0, \forall (i,j) \notin E\}.$$

Note that $\mathbf{K} \in \mathcal{K}(G)$ is positive semidefinite, and hence there exists $\mathbf{U} \in \mathbb{R}^{d \times n}$ such that $\mathbf{K} = \mathbf{U}^\top \mathbf{U}$. Note that $K_{ij} = \mathbf{u}_i^\top \mathbf{u}_j$ where $\mathbf{u}_i$ is the $i$-th column of $\mathbf{U}$. Hence by inspection it is clear that $\mathbf{U} \in Lab(G)$. Using a similar argument, we can show that for any $\mathbf{U} \in Lab(G)$, the matrix $\mathbf{K} = \mathbf{U}^\top \mathbf{U} \in \mathcal{K}(G)$. Thus, the two sets, $Lab(G)$ and $\mathcal{K}(G)$ are equivalent.

Furthermore, orthonormal embeddings are associated with an interesting quantity, the Lovász $\vartheta$ function [13, 7]. However, computing $\vartheta$ requires solving an SDP, which is impractical.

## 2 Generalization Bound for Graph Transduction using Orthonormal Embeddings

In this section we derive a generalization bound, used in the sequel for PAC analysis. We derive the following error bound, valid for any orthonormal embedding (supplementary material, Section B).

**Theorem 1** (Generalization bound). *Let $G = (V,E)$ be a simple graph with unknown binary labels $\mathbf{y} \in \mathcal{Y}^n$ on the vertices $V$. Let $\mathbf{K} \in \mathcal{K}(G)$. Given $G$, and labels of a randomly drawn subgraph $S$, let $\hat{\mathbf{y}} \in \widehat{\mathcal{Y}}^n$ be the predictions learnt by $\omega_C(\mathbf{K}, \mathbf{y}_S)$ in (3). Then, for $m \leq n/2$, with probability $\geq 1 - \delta$ over the choice of $S \subset V$, such that $|S| = m$*

$$er_{\bar{S}}^{0\text{-}1}[\hat{\mathbf{y}}] \leq \frac{1}{m} \sum_{i \in S} \ell^{hng}(y_i, \hat{y}_i) + 2C\sqrt{2\lambda_1(\mathbf{K})} + O\left(\sqrt{\frac{1}{m}\log\frac{1}{\delta}}\right). \tag{4}$$

Note that the above is a high-probability bound, in comparison to the expected analysis in (2). Also, the above result suggests that graph embeddings with low spectral norm and empirical error lead to better generalization. [1]'s analysis in (2) suggests that we should embed a graph on a unit sphere, however, does not help to choose the optimal embedding for graph transduction. Exploiting our analysis from (4), we present a spectral norm regularized algorithm in Section 3.

We would also like to study PAC learnability of orthonormal embeddings, where PAC learnability is defined as follows: given $G, \mathbf{y}$; does there exist an $\tilde{m} < n$, such that $w.p. \geq 1 - \delta$ over $S \subset V, |S| \geq \tilde{m}$; the generalization error $er_{\bar{S}}^{0\text{-}1} \leq \epsilon$. The quantity $\tilde{m}$ is termed as labelled sample complexity [2]. Existing analysis [2] do not apply to orthonormal embeddings as discussed in related work, Section 1. Theorem 1 allows us to derive improved statistical estimates (Section 3).

## 3 SPORE Formulation and PAC Analysis

Theorem 1 suggests that penalizing the spectral norm of $\mathbf{K}$ would lead to better generalization. To this end we motivate the following formulation.

$$\Psi_{C,\beta}(G, \mathbf{y}_S) = \min_{\mathbf{K} \in \mathcal{K}(G)} g(\mathbf{K}) \qquad \text{where} \quad g(\mathbf{K}) = \omega_C(\mathbf{K}, \mathbf{y}_S) + \beta\lambda_1(\mathbf{K}). \tag{5}$$

---

$^2(a)_+ = \max(a, 0) \; \forall a \in \mathbb{R}$

(5) gives an optimal orthonormal embedding, the optimal $\mathbf{K}$, which we will refer to as **SPORE**. In this section we first study the PAC learnability of SPORE, and derive a labelled sample complexity estimate. Next, we study efficient computation of SPORE. Though SPORE can be posed as an SDP, we show in Section 4 that it is possible to exploit the structure, and solve efficiently.

Given $G$ and $\mathbf{y}_S$, the function $\omega_C(\mathbf{K}, \mathbf{y}_S)$ is convex in $\mathbf{K}$ as it is the maximum of affine functions of $\mathbf{K}$. The spectral norm of $\mathbf{K}$, $\lambda_1(\mathbf{K})$ is also convex, and hence $g(\mathbf{K})$ is a convex function. Furthermore $\mathcal{K}(G)$ is an *Elliptope* [5], a convex body which can be described by the intersection of a positive semi-definite and affine constraints. It follows that hence (5) is convex. Usually these formulations are posed as SDPs which do not scale beyond few hundred vertices. In Section 4 we derive an efficient first order method which can solve for 1000's of vertices. Let $\mathbf{K}^*$ be the optimal embedding computed from (5). Note that once the kernel is fixed, the predictions are only dependent on $\omega_C(\mathbf{K}^*, \mathbf{y}_S)$. Let $\alpha^*$ be the solution to $\omega_C(\mathbf{K}^*, \mathbf{y}_S)$ as in (3), then the final predictions of (5) is given by $\hat{y}_i = \sum_{j \in S} K_{ij}^* \alpha_j^* y_j, \ \forall i \in [n]$.

At this point, we derive an interesting graph-dependent error convergence rate. We gather two important results, the proof of which appears in the supplementary material, Section C.

**Lemma 2.** *Given a simple graph $G = (V, E)$, $\max_{\mathbf{K} \in \mathcal{K}(G)} \lambda_1(\mathbf{K}) = \vartheta(\bar{G})$.*

**Lemma 3.** *Given $G$ and $\mathbf{y}$, for any $S \subseteq V$ and $C > 0$, $\min_{\mathbf{K} \in \mathcal{K}(G)} \omega_C(\mathbf{K}, \mathbf{y}_S) \leq \vartheta(G)/2$.*

In the standard PAC setting, there is a complete disconnection between the data distribution and target hypothesis. However, in the presence of unlabeled nodes, without any assumption on the data, it is impossible to learn labels. Following existing literature [1, 9], we work with *similarity graphs* – where presence of an edge would mean two nodes are similar; and derive the following (supplementary material, Section C).

**Theorem 4.** *Let $G = (V, E)$, $V = [n]$ be a simple graph with unknown binary labels $\mathbf{y} \in \mathcal{Y}^n$ on the vertices $V$. Given $G$, and labels of a randomly drawn subgraph $S \subset V$, $m = |S|$; let $\hat{\mathbf{y}}$ be the predictions learnt by SPORE (5), for parameters $C = \left( \frac{\vartheta(G)}{m\sqrt{\vartheta(\bar{G})}} \right)^{\frac{1}{2}}$ and $\beta = \frac{\vartheta(G)}{\vartheta(\bar{G})}$. Then, for $m \leq n/2$, with probability $\geq 1 - \delta$ over the choice of $S \subset V$, such that $|S| = m$*

$$er_{\bar{S}}^{0\text{-}1}[\hat{\mathbf{y}}] = O\Big( \frac{1}{m} \Big( \sqrt{n\vartheta(G)} + \log \frac{1}{\delta} \Big) \Big)^{\frac{1}{2}}. \tag{6}$$

*Proof.* (*Sketch*) Let $\mathbf{K}^*$ be the kernel learnt by SPORE (5). Using Theorem 1 and Lemma 2 for $\hat{\mathbf{y}}$

$$er_{\bar{S}}^{0\text{-}1}[\hat{\mathbf{y}}] \leq \frac{1}{m} \sum_{i \in S} \ell^{hng}(y_i, \hat{y}_i) + 2C\sqrt{2\vartheta(\bar{G})} + O\Big( \sqrt{\frac{1}{m} \log \frac{1}{\delta}} \Big). \tag{7}$$

From the primal formulation of (3), using Lemma 2 and 3, we get

$$C \sum_{i \in S} \ell^{hng}(y_i, \hat{y}_i) \leq \omega_C(\mathbf{K}^*, \mathbf{y}_S) \leq \Psi_{C,\beta}(G, \mathbf{y}_S) \leq \frac{\vartheta(G)}{2} + \beta\vartheta(\bar{G}).$$

Plugging back in (7), choosing $\beta$ such that $\frac{\beta}{Cm}\vartheta(\bar{G}) = 2C\sqrt{2\vartheta(\bar{G})}$ and optimizing for $C$ gives us the choice of parameters as stated. Finally, using $\vartheta(G)\vartheta(\bar{G}) = n$ [13] proves the result. $\square$

In the theorem above, $\bar{G}$ is the complement graph of $G$. The optimal orthonormal embedding $\mathbf{K}^*$ tend to embed vertices to nearby regions if they have connecting edges, hence, the notion of *similarity* is implicitly captured in the embedding. From (6), for a fixed $n$ and $m$, note that the error converges at a faster rate for a dense graph ($\vartheta$ is small), than for a sparse graph ($\vartheta$ is large). Such connections relating to graph structural properties were previously unavailable [1].

We also estimate the labelled sample complexity, by bounding (6) by $\epsilon > 0$, to obtain $\tilde{m} = \Omega\big(\frac{1}{\epsilon^2}(\sqrt{\vartheta n} + \log \frac{1}{\delta})\big)$. This connection helps to reason the intuition that for a sparse graph one would need a larger number of labelled vertices, than for a dense graph. For constants $\epsilon$, $\delta$, we obtain a fractional labelled sample complexity estimate of $\tilde{m}/n = \Omega\big(\vartheta/n\big)^{\frac{1}{2}}$, which is a significant improvement over the recently proposed $\Omega\big(\vartheta/n\big)^{\frac{1}{3}}$ [16]. The use of stronger machinery of

Rademacher averages (supplementary material, Section C), instead of VC-dimension [2], and specializing to SPORE allows us to improve over existing analysis [1, 16]. The proposed sample complexity estimate is interesting for $\vartheta = o(n)$, examples of such graphs include: **random graphs** ($\vartheta(G(n,p)) = \Theta(\sqrt{n})$) and **power-law graphs** ($\bar{\vartheta} = O(\sqrt{n})$).

# 4 Inexact Proximal methods for SPORE

In this section, we propose an efficient algorithm to solve SPORE (see (5)). The optimization problem SPORE can be posed as an SDP. Generic SDP solvers have a runtime complexity of $O(n^6)$ and often does not scale well for large graphs. We study first-order methods, such as projected subgradient procedures, as an alternative to SDPs, for minimizing $g(\mathbf{K})$. The main computational challenge in developing such procedures is that it is difficult to compute the projection on the *elliptope*. One could potentially use the seminal Dykstra's algorithm [3] of finding a feasible point in the intersection of two convex sets. The algorithm asymptotically finds a point in the intersection. This asymptotic convergence is a serious disadvantage in the usage of Dykstra's algorithm as a projection sub-routine. It would be useful to have an algorithm which after a finite number of iterations yield an approximate projection and a subsequent descent algorithm can yield a convergent algorithm.

Motivated by SPORE, we study the problem of minimizing non-smooth convex functions where the projection onto the feasible set can be computed only approximately. Recently there has been increasing interest in studying Inexact proximal methods [15, 18]. In the sequel we design an inexact proximal method which yields an $O(1/\sqrt{T})$ algorithm to solve (5). The algorithm is based on approximating the *prox* function by an iterative procedure which satisfies a suitably designed criterion.

## 4.1 An Infeasible Inexact Proximal (IIP) algorithm

Let $f$ be a convex function with properly defined sub-differential $\partial f(x)$ at every $x \in \mathcal{X}$. Consider the following optimization problem.

$$\min_{x \in \mathcal{X} \subset \mathbb{R}^d} f(x). \tag{8}$$

A subgradient projection iteration of the form

$$x_{k+1} = P_{\mathcal{X}}(x_k - \alpha_k h_k), \quad h_k \in \partial f(x_k) \tag{9}$$

is often used to arrive at an $\epsilon$ accurate solution by running the iterations $O(\frac{1}{\epsilon^2})$ number of times, where $P_{\mathcal{X}}(v)$ is the projection of $v \in \mathbb{R}^d$ on $\mathcal{X} \subset \mathbb{R}^d$ if $P_{\mathcal{X}}(v) = \operatorname{argmin}_{x \in \mathcal{X}} \frac{1}{2} \|v - x\|_F^2$. In many situations, such as $\mathcal{X} = \mathcal{K}(G)$, it is not possible to accurately compute the projection in finite amount of time and one may obtain only an approximate projection. Using the Moreau decomposition $P_{\mathcal{X}}(v) + Prox_{\sigma_{\mathcal{X}}}(v) = v$ [14], one can compute the projection if one could compute $\text{prox}_{\sigma_X}$, where $\sigma_A(a) = \max_{a \in \mathcal{X}} x^\top a$ is the support function of $\mathcal{X}$, and $\text{prox}_{\sigma_X}$ refers to the proximal operator for the function $g'$ at $v$ as defined below[3].

$$\text{prox}_{g'}(v) = \operatorname*{argmin}_{z \in \text{Dom}(g')} p_{g'}(z; v) \left( = \frac{1}{2} \|v - z\|^2 + g'(z) \right). \tag{10}$$

We assume that one could compute $z_{\mathcal{X}}^{\epsilon}(v)$, not necessarily in $\mathcal{X}$, such that

$$p_{\sigma_{\mathcal{X}}}(z_{\mathcal{X}}^{\epsilon}(v); v) \leq \min_{z \in \mathbb{R}^n} p_{\sigma_{\mathcal{X}}}(z; v) + \epsilon, \quad \text{and } P_{\mathcal{X}}^{\epsilon}(v) = v - z_{\mathcal{X}}^{\epsilon}. \tag{11}$$

See that $z_{\mathcal{X}}^{\epsilon}$ is an *inexact prox* and the resultant estimate of the projection $P_{\mathcal{X}}^{\epsilon}$ can be infeasible but hopefully not too far away. Note that $\epsilon = 0$ recovers the exact case. The next theorem confirms that it is possible to converge to the true optimum for a non-zero $\epsilon$ (supplementary material, Section D.5).

**Theorem 5.** *Consider the optimization problem* (8). *Starting from any* $\|x_0 - x^*\| \leq R$, *where* $x^*$ *is a solution of* (8), *for every k let us assume that we could obtain* $P_{\mathcal{X}}^{\epsilon}(y_k)$ *such that* $z_k = y_k - P_{\mathcal{X}}^{\epsilon}(y_k)$ *satisfy* (11), *where* $y_k = x_k - \alpha_k h_k$, $\alpha_k = \frac{s}{\|h_k\|}$, $\|h_k\| \leq L$, $\|x_k - x^*\| \leq R$, $s = \sqrt{\frac{R^2}{T} + \epsilon}$. *Then the iterates*

$$x_{k+1} = P_{\mathcal{X}}^{\epsilon}(x_k - \alpha_k h_k), \quad h_k \in \partial f(x_k) \tag{12}$$

$$yield \qquad f_T^* - f^* \leq L\sqrt{\frac{R^2}{T}} + \epsilon. \qquad (13)$$

**Related Work on Inexact Proximal methods:** There has been recent interest in deriving inexact proximal methods such as projected gradient descent, see [15, 18] for a comprehensive list of references. To the best of our knowledge, composite functions have been analyzed but no one has explored the case that $f$ is non-smooth. The results presented here are thus complementary to [15, 18]. Note the subtlety in using the proper approximation criteria. Using a distance criterion between the true projection and the approximate projection, or an approximate optimality criteria on the optimal distance would lead to a worse bound; using a dual approximate optimality criterion (here through the proximal operator for the support function) is key (as noted in [15, 18] and references therein).

As an immediate consequence of Theorem 5, note that suppose we have an algorithm to compute $\text{prox}_{\sigma_X}$ which guarantees after $S$ iterations that

$$p_{\sigma_X}(z_S; v) - \min_{z \in \mathbb{R}^d} p_{\sigma_X}(z; v) \leq \frac{\hat{R}^2}{S^2}, \qquad (14)$$

for a constant $\hat{R}$ particular to the set over which $p_{\sigma_X}$ is defined. We can initialize $\epsilon = \frac{\hat{R}^2}{S^2}$ in (13), that may suggest that one could use $S = \sqrt{T}$ iterations to yield

$$f_T^* - f^* \leq \frac{L\bar{R}}{\sqrt{T}} \quad \text{where} \quad \bar{R} = \sqrt{R^2 + \hat{R}^2}. \qquad (15)$$

**Remarks:** Computational efficiency dictates that the number of projection steps should be kept at a minimum. To this end we see that number of projection steps need to be at least $S = \sqrt{T}$ with the current choice of stepsizes. Let $c_p$ be the cost of one iteration of FISTA step and $c_0$ be the cost of one outer iteration. The total computation cost can be then estimated as $T^{3/2} \cdot c_p + T \cdot c_0$.

## 4.2 Applying IIP to compute SPORE

The problem of computing SPORE can be posed as minimizing a non-smooth convex function over an intersection of two sets: $\mathcal{K}(G) = \mathcal{S}_n^+ \cap P(G)$, intersection of positive semi-definite cone $\mathcal{S}_n^+$ and a *polytope* of equality constraints $P(G) := \{\mathbf{M} \in \mathcal{S}_n | M_{ii} = 1, M_{ij} = 0 \; \forall (i, j) \notin E\}$. The algorithm described in Theorem 5 readily applies to the new setting if the projection can be computed efficiently. The proximal operator for $\sigma_X$ can be derived as [4]

$$Prox_{\sigma_X}(v) = \operatorname*{argmin}_{a,b \in \mathbb{R}^d} p_{\sigma_X}(a, b; v) \left( = \frac{1}{2}\|(a + b) - v\|^2 + \sigma_A(a) + \sigma_B(b) \right). \qquad (16)$$

This means that even if we do not have an efficient procedure for computing $Prox_{\sigma_X}(v)$ directly, we can devise an algorithm to guarantee the approximation (11) if we can compute $Prox_{\sigma_A}(v)$ and $Prox_{\sigma_B}(v)$ efficiently. This can be done through the application of the popular FISTA algorithm for (16), which also guarantees (14). Algorithm 1 (detailed in the supplementary, named $IIP\_FISTA$), computes the following simple steps followed by the usual FISTA variable updates at each iteration $t$ : (a) gradient descent step on $a$ and $b$ with respect to the smooth term $\frac{1}{2}\|(a + b) - v\|^2$ and (b) proximal step with respect to $\sigma_A$ and $\sigma_B$ using the expressions (14), (21) (supplementary material).

Using the tools discussed above, we design Algorithm 1 to solve the SPORE formulation (5) using IIP. The proposed algorithm readily applies to general convex sets. However, we confine ourselves to specific sets of interest in our problem. The following theorem states the convergence rate of the proposed procedure.

**Theorem 6.** *Consider the optimization problem* (8) *with* $\mathcal{X} = A \bigcap B$, *where* $A$ *and* $B$ *are* $\mathcal{S}_n^+$ *and* $P(G)$ *respectively. Starting from any* $\mathbf{K}_0 \in A$ *the iterates* $\mathbf{K}_t$ *in Algorithm* (1) *satisfy*

$$\min_{t=0,\dots,T} f(\mathbf{K}_t) - f(\mathbf{K}^*) \leq \frac{L}{\sqrt{T}}\sqrt{R^2 + \hat{R}^2}.$$

*Proof.* Is an immediate extension of Theorem 5 – supplementary material, Section D.6. □

**Algorithm 1** IIP for SPORE
---
 1: **function** APPROX-PROJ-SUBG($\mathbf{K}_0, L, R, \hat{R}, T$)
 2: $\quad s = \frac{L}{\sqrt{T}} \cdot \left( \sqrt{R^2 + \hat{R}^2} \right)$          ▷ compute stepsize
 3: $\quad$ Initialize $t_0 = 1$.
 4: $\quad$ **for** $t = 1, \dots, T$ **do**
 5: $\quad\quad$ compute $h_{t-1}$      ▷ subgradient of $f(\mathbf{K})$ at $\mathbf{K}_{t-1}$ see equation (5)
 6: $\quad\quad v_t = \mathbf{K}_{t-1} - \frac{s}{\|h_{t-1}\|} h_{t-1}$
 7: $\quad\quad \tilde{\mathbf{K}}_t = IIP\_FISTA(v_t, \sqrt{T})$     ▷ FISTA for $\sqrt{T}$ steps. Use Algorithm 1 (supp.)
 8: $\quad\quad \mathbf{K}_t = Proj_A(\tilde{\mathbf{K}}_t) = \mathbf{K}_t - \text{prox}_{\sigma_A}(\mathbf{K}_t)$
 9: $\quad\quad$                 ▷ $\mathbf{K}_t$ needs to be psd for the next SVM call. Use (14) (supp.)
10: $\quad$ **end for**
11: **end function**
---

Equating the problem (8) with the SPORE problem (5), we have $f(\mathbf{K}) = \omega_C(\mathbf{K}, \mathbf{y}_S) + \beta \lambda_1(\mathbf{K})$. The set of subgradients of $f$ at the iteration $t$ is given by $\partial f(\mathbf{K}_t) = \{ -\frac{1}{2} \mathbf{Y} \alpha_t \alpha_t^\top \mathbf{Y} + \beta \mathbf{v}_t \mathbf{v}_t^\top | \alpha_t$ is returned by SVM, and $\mathbf{v}_t$ is the eigen vector corresponding to $\lambda_1(K_t) \}^5$, where $\mathbf{Y}$ be a diagonal matrix such that $Y_{ii} = y_i$, for $i \in S$, and $0$ otherwise. The step size is calculated using estimates of $L, R$ and $\hat{R}$, which can be derived as $L = nC^2, R = n, \hat{R} = n^{2.5}$ for the SPORE problem. Check the supplementary material for the derivations.

## 5  Multiple Graph Transduction

Multiple graph transduction is of recent interest in a multi-view setting, where individual views are expressed by a graph. This includes many practical problems in bioinformatics [17], spam detection [21], etc. We propose an MKL style extension of SPORE, with improved PAC bounds.

Formally, the problem of multiple graph transduction is stated as – let $\mathbb{G} = \{G^{(1)}, \dots, G^{(M)}\}$ be a set of simple graphs $G^{(k)} = (V, E^{(k)})$, defined on a common vertex set $V = [n]$. Given $\mathbb{G}$ and $\mathbf{y}_S$ as before, the goal is to accurately predict $\mathbf{y}_{\bar{S}}$. Following the standard technique of taking convex combination of graph kernels [16], we propose the following **MKL-SPORE** formulation

$$\Phi_{C,\beta}(\mathbb{G}, \mathbf{y}_S) = \min_{\mathbf{K}^{(k)} \in \mathcal{K}(G^{(k)})} \left( \min_{\eta \in \mathcal{S}^{M-1}} \omega_C \big( \sum_{k \in [M]} \eta_k \mathbf{K}^{(k)}, \mathbf{y}_S \big) + \beta \max_{k \in [M]} \lambda_1(\mathbf{K}^{(k)}) \right). \quad (17)$$

Similar to Theorem 4, we can show the following (supplementary material, Theorem 8)

$$er_{\bar{S}}^{0\text{-}1}[\hat{\mathbf{y}}] = O\Big( \frac{1}{m} \Big( \sqrt{n\vartheta(\mathbb{G})} + \log \frac{1}{\delta} \Big) \Big)^{\frac{1}{2}} \quad where \quad \vartheta(\mathbb{G}) \leq \min_{k \in [M]} \vartheta(G^{(k)}). \quad (18)$$

It immediately follows that combining multiple graphs improves the error convergence rate (see (6)), and hence the labelled sample complexity. Also, the bound suggests that the presence of at least one "good" graph is sufficient for MKL-SPORE to learn accurate predictions. This motivates us to use the proposed formulation in the presence of noisy graphs (Section 6). We can also apply the IIP algorithm described in Section 4 to solve for (17) (supplementary material, Section F).

## 6  Experiments

We conducted experiments on both real world and synthetic graphs, to illustrate our theoretical observations. All experiments were run on CPU with 2 Xeon Quad-Core processors (2.66GHz, 12MB L2 Cache) and 16GB memory running CentOS 5.3.

---
$^5 \alpha_t = \text{argmax}_{\alpha \in \mathbb{R}_+^n, \|\alpha\|_\infty \leq C \atop \alpha_j = 0 \ \forall j \notin S} \alpha^\top \mathbf{1} - \frac{1}{2}\alpha^\top \mathbf{Y} \mathbf{K}_t \mathbf{Y} \alpha$ and $\mathbf{v}_t = \text{argmax}_{\mathbf{v} \in \mathbb{R}^n, \|v\|=1} \mathbf{v}^\top \mathbf{K}_t \mathbf{v}$

Table 1: SPORE comparison.

| Dataset | Un-Lap | N-Lap | KS | SPORE |
|---|---|---|---|---|
| *breast-cancer* | 88.22 | 93.33 | 92.77 | **96.67** |
| *diabetes* | 68.89 | 69.33 | 69.44 | **73.33** |
| *fourclass* | 70.00 | 70.00 | 70.44 | **78.00** |
| *heart* | 71.97 | 75.56 | 76.42 | **81.97** |
| *ionosphere* | 67.77 | 68.00 | 68.11 | **76.11** |
| *sonar* | 58.81 | 58.97 | 59.29 | **63.92** |
| *mnist-1vs2* | 75.55 | 80.55 | 79.66 | **85.77** |
| *mnist-3v8* | 76.88 | 81.88 | 83.33 | **86.11** |
| *mnist-4v9* | 68.44 | 72.00 | 72.22 | **74.88** |

Table 2: Large Scale – 2000 Nodes.

| Dataset | Un-Lap | N-Lap | KS | SPORE |
|---|---|---|---|---|
| *mnist-1vs2* | 83.80 | 96.23 | 94.95 | **96.72** |
| *mnist-3vs8* | 55.15 | 87.35 | 87.35 | **91.35** |
| *mnist-5vs6* | 96.30 | 94.90 | 92.05 | **97.35** |
| *mnist-1vs7* | 90.65 | 96.80 | 96.55 | **97.25** |
| *mnist-4vs9* | 65.55 | 65.05 | 61.30 | **87.40** |

**Graph Transduction (SPORE):**  We use two datasets UCI [12] and MNIST [10]. For the UCI datasets, we use the RBF kernel[6] and threshold with the mean, and for the MNIST datasets we construct a similarity matrix using cosine distance for a random sample of 500 nodes, and threshold with 0.4 to obtain unweighted graphs. With 10% labelled nodes, we compare SPORE with formulation (3) using graph kernels – Unnormalized Laplacian $(c_1 \mathbf{I} + \mathbf{L})^{-1}$, Normalized Laplacian $\left(c_2 \mathbf{I} + \mathbf{D}^{-\frac{1}{2}} \mathbf{L} \mathbf{D}^{-\frac{1}{2}}\right)^{-1}$ and **K**-Scaling [1], where $\mathbf{L} = \mathbf{D} - \mathbf{A}$, $\mathbf{D}$ being a diagonal matrix of degrees. We choose parameters $c_1$, $c_2$, $C$ and $\beta$ by cross validation. Table 1 summarizes the results, averaged over 5 different labelled samples, with each entry being accuracy in % *w.r.t.* 0-1 loss function. As expected from Section 3, SPORE significantly outperforms existing methods. We also tackle large scale graph transduction problems, Table 2 shows superior performance of Algorithm 1 for a random sample of 2000 nodes, with only 5 outer iterations and 20 inner projections.

**Multiple Graph Transduction (MKL-SPORE):**  We illustrate the effectiveness of combining multiple graphs, using mixture of random graphs – $G(p, q)$, $p, q \in [0, 1]$ where we fix $|V| = n = 100$ and the labels $\mathbf{y} \in \mathcal{Y}^n$ such that $y_i = 1$ if $i \leq n/2$; $-1$ otherwise. An edge $(i, j)$ is present with probability $p$ if $y_i = y_j$; otherwise present with probability $q$. We generate three datasets to simulate homogenous, heterogenous and noisy case, shown in Table 3.

Table 4: Superior performance of MKL-SPORE.

| Graph | Homo. | Heter. | Noisy |
|---|---|---|---|
| $G^{(1)}$ | 84.4 | 69.2 | **84.4** |
| $G^{(2)}$ | 84.8 | 68.6 | 68.2 |
| $G^{(3)}$ | 86.4 | 72.0 | 54.4 |
| Union | 85.5 | 69.3 | 69.3 |
| Intersection | 83.8 | 67.5 | 69.0 |
| Majority | 93.7 | 76.9 | 76.6 |
| Multiple Graphs | **95.6** | **80.6** | 81.9 |

Table 3: Synthetic multiple graphs dataset.

| Graph | Homo. | Heter. | Noisy |
|---|---|---|---|
| $G^{(1)}$ | $G(0.7, 0.3)$ | $G(0.7, 0.5)$ | $G(0.7, 0.3)$ |
| $G^{(2)}$ | $G(0.7, 0.3)$ | $G(0.6, 0.4)$ | $G(0.6, 0.4)$ |
| $G^{(3)}$ | $G(0.7, 0.3)$ | $G(0.5, 0.3)$ | $G(0.5, 0.5)$ |

MKL-SPORE was compared with individual graphs; and with the union, intersection and majority graphs[7]. We use SPORE to solve for single graph transduction, and the results were averaged over 10 random samples of 5% labelled nodes. For the comparison metric as before, Table 4 shows that combining multiple graphs improves classification accuracy. Furthermore, the noisy case illustrates the robustness of the proposed formulation, a key observation from (18).

## 7   Conclusion

We show that the class of orthonormal graph embeddings are efficiently PAC learnable. Our analysis motivates a Spectral Norm regularized formulation – SPORE for graph transduction. Using inexact proximal method, we design an efficient first order method to solve for the proposed formulation. The algorithm and analysis presented readily generalize to the multiple graphs setting.

**Acknowledgments**

We acknowledge support from a grant from Indo-French Center for Applied Mathematics (IFCAM).

## Footnotes

[1]A simple graph is an unweighted, undirected graph with no self loops or multiple edges.

[3]A more general definition of the proximal operator is $- \text{prox}_{g'}^{\tau}(v) = \operatorname{argmin}_{z \in \text{Dom}(g')} \frac{1}{2\tau} \|v - z\|^2 + g'(z)$

[4] The derivation is presented in supplementary material, Claim 6.

[6]The $(i, j)^{th}$ entry of an RBF kernel is given by $\exp\left(\frac{\|\mathbf{x}_i - \mathbf{x}_j\|^2}{2\sigma^2}\right)$, where $\sigma$ is set as the mean distance.

[7]Majority graph is a graph where an edge $(i, j)$ is present, if a majority of the graphs have the edge $(i, j)$.

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
