[Supplementary Material · supp.pdf]

# Spectral Norm Regularization of Orthonormal Representations for Graph Transduction

**Rakesh Shivanna**
Google Inc.
Mountain View, CA, USA
rakeshshivanna@google.com

**Bibaswan Chatterjee**
Dept. of Computer Science & Automation
Indian Institute of Science, Bangalore
bibaswan.chatterjee@csa.iisc.ernet.in

**Raman Sankaran, Chiranjib Bhattacharyya**
Dept. of Computer Science & Automation
Indian Institute of Science, Bangalore
ramans,chiru@csa.iisc.ernet.in

**Francis Bach**
INRIA - Sierra Project-team
École Normale Supérieure, Paris, France
francis.bach@ens.fr

## A  Preliminaries and Definitions

For completeness, we state/prove some of the non-trivial results used in the paper.

**Notations.** Let $\| \cdot \|_0$ and $\| \cdot \|_\infty$ be the L0 and L-infinity norms respectively. For $y \in \mathcal{Y}$ and $\hat{y} \in \widehat{\mathcal{Y}}$; let $\ell^{hng}(y,\hat{y}) = (1 - y\hat{y})_+$ [1], $\ell^r(y,\hat{y}) = \min\{1, (1-y\hat{y})_+\}$ and $\ell^{0\text{-}1}(y,\hat{y}) = \mathbb{1}[y\hat{y} < 0]$ denote the hinge, ramp and 0-1 loss respectively. Note that these loss functions upper bound the other in same order, $\ell^{hng} \geq \ell^r \geq \ell^{0\text{-}1}$.

**Asymptotic Notations [1].** For non-negative functions $f_1(n)$ and $f_2(n)$

- $f_1(n) = O(f_2(n)) \implies \exists n_0$ and a constant $c > 0$ such that $\forall n > n_0$, $f_1(n) \leq cf_2(n)$.
- $f_1(n) = \Omega(f_2(n)) \implies \exists n_0$ and a constant $c > 0$ such that $\forall n > n_0$, $f_1(n) \geq cf_2(n)$.
- $f_1(n) = \Theta(f_2(n))$ iff $f_1(n) = O(f_2(n))$ and $f_1(n) = \Omega(f_2(n))$.

## B  Generalization Bound

Let $\mathbf{U} \in Lab(G)$ be the orthonormal embedding corresponding to the graph kernel $\mathbf{K} \in \mathcal{K}(G)$. Note that the classifier learnt by the SVM formulation $\omega_C(\mathbf{K}, \mathbf{y}_S)$ as in (3) (paper), is of the form $h = \mathbf{U}\alpha$, where $\alpha$ is in the feasible set. In general, we define the following function class associated with the orthonormal embedding

$$\tilde{\mathcal{H}}_{\mathbf{U}} = \left\{ h \big| h = \mathbf{U}\alpha, \alpha \in \mathbb{R}^n, \|\alpha\|_\infty \leq C, \|\alpha\|_0 \leq m \right\} \tag{1}$$

We follow a similar proof technique as in [3], however specialize to the class of orthonormal representation and SPORE formulation.

**Theorem 1.** (*Paper*) *Let $G = (V, E)$ be a simple graph with unknown binary labels $\mathbf{y} \in \mathcal{Y}^n$ on the vertices $V$. Let $\mathbf{K} \in \mathcal{K}(G)$. Given $G$, and labels of a randomly drawn subgraph $S$, let $\hat{\mathbf{y}} \in \widehat{\mathcal{Y}}^n$ be the predictions learnt by $\omega_C(\mathbf{K}, \mathbf{y}_S)$ as in (3) (paper). Then, for $m \leq n/2$, with probability $\geq 1 - \delta$ over the choice of $S \subset V$, such that $|S| = m$*

$$er_{\bar{S}}^{0\text{-}1}[\hat{\mathbf{y}}] \leq \frac{1}{m} \sum_{i \in S} \ell^{hng}(y_i, \hat{y}_i) + 2C\sqrt{2\lambda_1(\mathbf{K})} + O\left( \sqrt{\frac{1}{m} \log \frac{1}{\delta}} \right) \tag{2}$$

*Proof.* Let $\mathbf{U} \in Lab(G)$ be the orthonormal representation associated with $\mathbf{K}$. Let $\pi = [\pi_1, \ldots, \pi_n]$ denote a permutation on $[n]$. For any $\pi$, without loss of generality, let the first $m \in [n]$ nodes be labelled. Let $er_S^{r;\pi}[\hat{\mathbf{y}}] = \frac{1}{m} \sum_{i=1}^{m} \ell^r(y_{\pi_i}, \hat{y}_{\pi_i})$, where $\ell^r$ is the ramp loss as in Section A; and similarly for $er_{\bar{S}}^{r;\pi}[\hat{\mathbf{y}}]$. Let $\tilde{\pi} = [1, 2, \ldots, n]$ denote the trivial permutation. Introduce a ghost permutation –

$$er_{\bar{S}}^{r,\tilde{\pi}}[\hat{\mathbf{y}}] \leq er_S^{r;\tilde{\pi}}[\hat{\mathbf{y}}] + er_{\bar{S}}^{r;\tilde{\pi}}[\hat{\mathbf{y}}] - er_S^{r;\tilde{\pi}}[\hat{\mathbf{y}}]$$

$$\leq er_S^{r,\tilde{\pi}}[\hat{\mathbf{y}}] + \sup_{\tilde{\mathbf{y}} \in \tilde{\mathcal{Y}}_{\mathbf{U}}} \left[ er_{\bar{S}}^{r,\tilde{\pi}}[\tilde{\mathbf{y}}] - er_S^{r,\tilde{\pi}}[\tilde{\mathbf{y}}] \right] \leq er_S^{r,\tilde{\pi}}[\hat{\mathbf{y}}] + \Phi_{\mathbf{U}}^{\tilde{\pi}}[\hat{\mathbf{y}}] \qquad (3)$$

where

$$\Phi_{\mathbf{U}}^{\tilde{\pi}}[\hat{\mathbf{y}}] = \mathbb{E}_\pi \sup_{\tilde{\mathbf{y}} \in \tilde{\mathcal{Y}}_{\mathbf{U}}} \left[ er_{\bar{S}}^{r,\tilde{\pi}}[\hat{\mathbf{y}}] - er_{\bar{S}}^{r;\pi}[\tilde{\mathbf{y}}] + er_S^{r;\pi}[\tilde{\mathbf{y}}] - er_S^{r,\tilde{\pi}}[\hat{\mathbf{y}}] \right]$$

where $\tilde{\mathcal{Y}}_{\mathbf{U}} = \{\tilde{\mathbf{y}} | \tilde{\mathbf{y}} = \mathbf{U}^\top h, h \in \tilde{\mathcal{H}}_{\mathbf{U}}\}$, for $\tilde{\mathcal{H}}_{\mathbf{U}}$ as in (1). (3) follows by adding and subtracting $\mathbb{E}_\pi \left[ er_{\bar{S}}^{r;\pi}[\hat{\mathbf{y}}] \right] = \mathbb{E}_\pi \left[ er_S^{r;\pi}[\hat{\mathbf{y}}] \right]$ inside the supremum and applying Jensen's inequality to bring the expectation of out the supremum.

We use Doob's martingale process (see [2]) to bound the function $\Phi_{\mathbf{U}}^{\tilde{\pi}}$ by its expectation –

**Lemma 1.** *Given $n \in \mathbb{N}$ and $m \in [n]$, let $\pi$ be a random permutation vertor over $[n]$. Let $\pi^{ij}$ to denote perturbed permutation, where $i^{th}$ and $j^{th}$ elements are exchanged. Let $f(\pi)$ be an $\pi$-permutation symmetric function satisfying $|f(\pi) - f(\pi^{ij})| \leq \beta \; \forall i \in [m], \; j \notin [m]$. Then*

$$\mathbf{Pr}_\pi \{ f(\pi) - \mathbb{E}_\pi[f(\pi)] \geq \epsilon \} \leq \exp \left( -\frac{\epsilon^2}{\beta^2 m} \right)$$

Using the above for $\beta = \frac{2}{m}$, we get for $\delta > 0$, *w.p.* $\geq 1 - \delta$ over the permutation $\tilde{\pi}$

$$\Phi_{\mathbf{U}}^{\tilde{\pi}}[\hat{\mathbf{y}}] \leq \mathbb{E}_\pi \left[ \Phi_{\mathbf{U}}^{\pi}[\hat{\mathbf{y}}] \right] + 2 \sqrt{\frac{1}{m} \log \frac{1}{\delta}} \qquad (4)$$

Now we bound $\mathbb{E}_\pi \left[ \Phi_{\mathbf{U}}^{\pi}[\hat{\mathbf{y}}] \right]$ using results of [3] as follows

$$\mathbb{E}_\pi \left[ \Phi^\pi[\hat{\mathbf{y}}] \right] \leq R \left( \mathcal{L}_{\mathbf{U}}^r, m, \frac{mu}{n^2} \right) + O \left( \frac{1}{\sqrt{m}} \right) \qquad (5)$$

where $\mathcal{L}_{\mathbf{U}}^r = \{ [\ell^r(y_1, \tilde{y}_1), \ldots, \ell^r(y_n, \tilde{y}_n)] | \tilde{\mathbf{y}} \in \tilde{\mathcal{Y}}_{\mathbf{U}} \}$, and $R(\mathcal{V}, m) := \frac{2}{m} \mathbb{E}_\sigma \left[ \sup_{\mathbf{v} \in \mathcal{V}} \mathbf{v}^\top \sigma \right]$ is the *Rademacher average* of the vector space $\mathcal{V} \subseteq \mathbb{R}^n$, where $\sigma$ is an *i.i.d.* random vector, each entry taking values $+1, \; -1, \; 0$ with probability $p, \; p, \; 1 - p$ respectively; $p := m/n$. We recall the following interesting property of Rademacher averages [3] –

**Lemma 2.** *For any $\mathcal{V} \subseteq \mathbb{R}^n$, $m \in [n]$ and $0 \leq p_1 \leq p_2 \leq \frac{1}{2}$, $R(\mathcal{V}, m, p_1) \leq R(\mathcal{V}, m, p_2)$.*

Using the above, we get

$$R \left( \mathcal{L}_{\mathbf{U}}^r, m, \frac{mu}{n^2} \right) \leq R \left( \mathcal{L}_{\mathbf{U}}^r, m, \frac{m}{n} \right) \leq R \left( \tilde{\mathcal{Y}}_{\mathbf{U}}, m, \frac{m}{n} \right) \qquad (6)$$

where the last inequality follows from the contraction property [3] of the class complexity. Note that (6) relates to the function class, for which we can derive a tight estimate as follows – We derive the following tight Rademacher complexity estimate for the class of orthonormal embeddings –

**Lemma 3.** $R \left( \tilde{\mathcal{Y}}_{\mathbf{U}}, m, \frac{m}{n} \right) \leq 2C \sqrt{2\lambda_1(\mathbf{K})}$

For any random vector $\sigma$, the supremum is given by

$$\sup_{\tilde{\mathbf{y}} \in \tilde{\mathcal{Y}}_{\mathbf{U}}} \sum_{i \in [n]} \sigma_i \tilde{y}_i = \sup_{h \in \tilde{\mathcal{H}}_{\mathbf{U}}} \sum_{i \in [n]} \sigma_i \langle h, \mathbf{u}_i \rangle = \sup_{h \in \tilde{\mathcal{H}}_{\mathbf{U}}} \left\langle h, \sum_{i \in [n]} \sigma_i \mathbf{u}_i \right\rangle \leq C \sqrt{m\lambda_1(\mathbf{K})} \left\| \sum_{i \in [n]} \sigma_i \mathbf{u}_i \right\|$$

The last equality from optimality over supremum and the norm constraint

$$\max_{h \in \tilde{\mathcal{H}}_{\mathbf{U}}} \|h\| = \max_{\|\alpha\|_\infty \leq C, \|\alpha\|_0 \leq m} \sqrt{\alpha^\top \mathbf{K} \alpha} \leq C \sqrt{m \lambda_1(\mathbf{K})}$$

Now, taking expectation over $\sigma$, one obtains

$$R\left(\tilde{\mathcal{Y}}_{\mathbf{U}}, m, \frac{m}{n}\right) = 2C \sqrt{\frac{\lambda_1(\mathbf{K})}{m}} \mathbb{E}_\sigma\left[\sqrt{\sigma^\top \mathbf{K} \sigma}\right] \tag{7}$$

Using Jensen's inequality, the expectation term can be upper bounded by $\sqrt{\mathbb{E}_\sigma[\sigma^\top \mathbf{K} \sigma]}$. Further, using *i.i.d.* assumption of $\sigma$, the expectation evaluates to $(2m/n) \sum_{i \in [n]} K_{ii} = 2m$. Plugging back in (7) proves the lemma. Finally, (2) is immediate by combining Lemma 3, (6), (5), (4) and (3). $\square$

## C   SPORE formulation and PAC analysis

We show that the spectral norm of the kernel relates to the structural properties of the graph

**Lemma 2.** (*Paper*) *Given a simple, undirected graph* $G = (V, E)$, $\max_{\mathbf{K} \in \mathcal{K}(G)} \lambda_1(\mathbf{K}) = \vartheta(\bar{G})$.

*Proof.* We recall another definition of the $\vartheta$ function [6]

$$\vartheta(\bar{G}) = \max_{\mathbf{U} \in Lab(G)} \max_{\mathbf{c} \in \mathcal{S}_2^{d-1}} \sum_{i \in [n]} \left(\mathbf{u}_i^\top \mathbf{c}\right)^2 \tag{8}$$

For a fixed $\mathbf{U} \in Lab(G)$ and $\mathbf{c} \in \mathcal{S}_2^{d-1}$, the summation evaluates to $\mathbf{c}^\top \mathbf{U}\mathbf{U}^\top \mathbf{c}$. Thus, for any fixed $\mathbf{U} \in Lab(G)$, $\max_{\mathbf{c} \in \mathcal{S}_2^{d-1}} \sum_{i \in [n]} \left(\mathbf{u}_i^\top \mathbf{c}\right)^2 = \lambda_1(\mathbf{U}\mathbf{U}^\top)$. From first principles, $\lambda_1(\mathbf{U}\mathbf{U}^\top) = \lambda_1(\mathbf{U}^\top\mathbf{U}) = \lambda_1(\mathbf{K})$, where $\mathbf{K} = \mathbf{U}^\top\mathbf{U} \in \mathcal{K}(G)$ (Section 1, paper). As there is correspondence between the two sets $Lab(G)$ and $\mathcal{K}(G)$ (Section 1), (8) evaluates to $\vartheta(\bar{G}) = \max_{\mathbf{K} \in \mathcal{K}(G)} \lambda_1(\mathbf{K})$, thus proving the claim. $\square$

Following the proof technique of [7] and [5], we prove –

**Lemma 3.** (*paper*) *Given $G$ and $\mathbf{y}$, for any $S \subseteq V$ and $C > 0$*

$$\min_{\mathbf{K} \in \mathcal{K}(G)} \omega_C(\mathbf{K}^S, \mathbf{y}_S) \leq \vartheta(G)/2$$

*Proof.* We recall another definition of the $\vartheta$ function [6]

$$\vartheta(G) = \min_{\mathbf{U} \in Lab(G)} \min_{\mathbf{c} \in \mathcal{S}_2^{d-1}} \max_{i \in [n]} \frac{1}{\left(\mathbf{c}^\top \mathbf{u}_i\right)^2} \tag{9}$$

For a fixed $\mathbf{K}$, from the primal of SVM formulation, it follows that

$$\omega_\infty(\mathbf{K}^S, \mathbf{y}_S) = \min_{\mathbf{w} \in \mathbb{R}^d} \frac{1}{2}\|\mathbf{w}\|^2 \quad \text{s.t.} \quad y_i \mathbf{w}^\top \mathbf{u}_i \geq 1 \, \forall i \in S$$

where $\mathbf{U}$ is the orthonormal representation corresponding to $\mathbf{K}$ (Section 1). Now, writing $\mathbf{w} = t\mathbf{c}$, where $\mathbf{c} \in \mathcal{S}_2^{d-1}$

$$= \min_{t, \mathbf{c} \in \mathcal{S}_2^{d-1}} t^2 \quad \text{s.t.} \quad t \geq \frac{1}{y_i \mathbf{c}^\top \mathbf{u}_i} \, \forall i \in S$$

$$= \min_{\mathbf{c} \in \mathcal{S}_2^{d-1}} \max_{i \in S} \frac{1}{(y_i \mathbf{c}^\top \mathbf{u}_i)^2}$$

$$= \min_{\mathbf{c} \in \mathcal{S}_2^{d-1}} \max_{i \in S} \frac{1}{(\mathbf{c}^\top \mathbf{u}_i)^2} \leq \min_{\mathbf{c} \in \mathcal{S}_2^{d-1}} \max_{i \in [n]} \frac{1}{(\mathbf{c}^\top \mathbf{u}_i)^2}$$

Thus, the proof follows from (9), using a trivial bound $\omega_C(\mathbf{K}^S, \mathbf{y}_S) \leq \omega_\infty(\mathbf{K}^S, \mathbf{y}_S)$ and noting that the sets $Lab(G)$ and $\mathcal{K}(G)$ are equivalent Section 1. $\square$

**Theorem 4.** (*paper*) *Let* $G = (V, E)$, $V = [n]$ *be a simple graph with unknown binary labels* $\mathbf{y} \in \mathcal{Y}^n$ *on the vertices* $V$. *Given* $G$, *and labels of a randomly drawn subgraph* $S \subset V$, $m = |S|$; *let* $\hat{\mathbf{y}}$ *be the predictions learnt by SPORE* (5), *for parameters* $C = \left( \frac{\vartheta(G)}{m\sqrt{\vartheta(\bar{G})}} \right)^{\frac{1}{2}}$ *and* $\beta = \frac{\vartheta(G)}{\vartheta(\bar{G})}$. *Then, for* $m \leq n/2$, *with probability* $\geq 1 - \delta$ *over the choice of* $S \subset V$, *such that* $|S| = m$

$$er_S^{0\text{-}1}[\hat{\mathbf{y}}] = O\left( \frac{1}{m} \left( \sqrt{n\vartheta(G)} + \log \frac{1}{\delta} \right) \right)^{\frac{1}{2}}$$

*Proof.* Let $\mathbf{K}^*$ be the kernel learnt by SPORE (5). Using Theorem 1 (paper) for final predictions $\hat{\mathbf{y}}$, we obtain

$$er_S^{0\text{-}1}[\hat{\mathbf{y}}] \leq \frac{1}{m} \sum_{i \in S} \ell^{hng}(y_i, \hat{y}_i) + 2C\sqrt{2\lambda(\mathbf{K}^*)} + O\left( \sqrt{\frac{1}{m} \log \frac{1}{\delta}} \right)$$

Using Lemma 2 (paper) $\lambda(\mathbf{K}^*) \leq \vartheta(\bar{G})$, where $\bar{G}$ is complement graph of $G$

$$\leq \frac{1}{m} \sum_{i \in S} \ell^{hng}(y_i, \hat{y}_i) + 2C\sqrt{2\vartheta(\bar{G})} + O\left( \sqrt{\frac{1}{m} \log \frac{1}{\delta}} \right) \tag{10}$$

Recall the primal formulation of (3) (paper) for $\mathbf{K}^* = \mathbf{U}^{*\top}\mathbf{U}^*$, $\mathbf{U}^* \in Lab(G)$ (Section 1)

$$\omega_C(\mathbf{K}^*, \mathbf{y}_S) = \min_{\mathbf{w} \in \mathbb{R}^d} \frac{1}{2} \|\mathbf{w}\|_2^2 + C \sum_{i \in S} \ell^{hng}(y_i, \mathbf{w}^\top \mathbf{u}_i^*)$$

Let $\mathbf{w}^*$ be the solution at optimal, then note that $\hat{y}_i = \mathbf{w}^{*\top}\mathbf{u}_i^*$, $\forall i \in [n]$. Thus, we bound the empirical error as follows

$$C \sum_{i \in S} \ell^{hng}(y_i, \hat{y}_i) = \omega_C(\mathbf{K}^*, \mathbf{y}_S) - \frac{1}{2}\|\mathbf{w}\|^2 \leq \omega_C(\mathbf{K}^*, \mathbf{y}_S)$$

$$\leq \Psi_{C,\beta}(G, \mathbf{y}_S) = \min_{\mathbf{K} \in \mathcal{K}(G)} \omega_C(\mathbf{K}^S, \mathbf{y}_S) + \beta \lambda_1(\mathbf{K})$$

$$\leq \min_{\mathbf{K} \in \mathcal{K}(G)} \omega_C(\mathbf{K}^S, \mathbf{y}_S) + \beta \max_{\mathbf{K} \in \mathcal{K}(G)} \lambda_1(\mathbf{K}) \leq \frac{\vartheta(G)}{2} + \beta\vartheta(\bar{G})$$

The last inequality follows from Lemma 2 and 3 (paper). Plugging back in (10), we get

$$er_S^{0\text{-}1}[\hat{\mathbf{y}}] \leq \frac{1}{2Cm}\vartheta(G) + \frac{\beta}{Cm}\vartheta(\bar{G}) + 2C\sqrt{2\vartheta(\bar{G})} + O\left( \sqrt{\frac{1}{m} \log \frac{1}{\delta}} \right) \tag{11}$$

Choosing $\beta$ such that $\frac{\beta}{Cm}\vartheta(\bar{G}) = 2C\sqrt{2\vartheta(\bar{G})}$ and optimizing for $C$ gives us the choice of parameters as in the statement of the theorem. Plugging back in (11), we get

$$= O\left( \frac{1}{\sqrt{m}} \left( \sqrt{\vartheta(G)\sqrt{\vartheta(\bar{G})}} + \sqrt{\log \frac{1}{\delta}} \right) \right)$$

Finally, using $\vartheta(G)\vartheta(\bar{G}) = n$ [6], and concavity $\sqrt{a} + \sqrt{b} \leq \sqrt{2(a+b)}$ proves the result. $\square$

# D  Proposed Algorithms

## D.1  The Projection Algorithm

Algorithm 1 lists the steps of the accelerated gradient descent algorithm FISTA applied to 16. The objective $p_{\sigma_\mathcal{X}}$ is composed of the smooth term $p_{\sigma_\mathcal{X}}^s(a, b) = \frac{1}{2}\|v - a - b\|^2$ with the Lipshitz continuous gradients with constant $L_p = 1$, and the non smooth term $p_{\sigma_\mathcal{X}}^n(a, b) = \sigma_A(a) + \sigma_B(b)$. Step 6 executes a gradient descent on $a$ with respect to $p_{\sigma_\mathcal{X}}$ followed by a proximal mapping with $\sigma_A$. The gradient of $p_{\sigma_\mathcal{X}}$ equals $a + b - v$, and the stepsize chosen to be $\frac{1}{L_p} = 1$. Similarly step 6 perform the

gradient descent on $b$ followed by the proximal mapping with $\sigma_B$, The definitions for the support functions and its associated operators are provided in Section D.3.

The algorithm requires as input the current iterate $v_k = x_k - \alpha_k g_k$ to project into the set $\mathcal{X}$, and the number of iterations $S$ and returns an approximate projection $x_{k+1} = v_k - a_S - b_S$.

For a graph $G = (V, E)$, the set $\mathcal{X} = A \bigcap B$ is defined with

$$A = \{\mathbf{K} \in \mathbb{R}^{n \times n} | \mathbf{K} \succeq 0, tr(\mathbf{K}) = n\} \text{ and } B = \{\mathbf{K} \in \mathbb{R}^{n \times n} | \mathbf{K}_{ii} = 1, i \in [n], \mathbf{K}_{ij} = 0, (i, j) \notin E\} \tag{12}$$

---

**Algorithm 1** Accelerated Gradient Descent (FISTA) to solve 11

---

1: **function** $IIP\ FISTA(v, S)$
2:     Initialize $a_0 = 0, b_0 = 0$.
3:     Initialize $(\hat{a}_0, \hat{b}_0) = (a_0, b_0)$.
4:     Initialize $t_0 = 1$.
5:     **for** $t = 1, \ldots, S$ **do**
6:         $a_t = \text{prox}_{\sigma_A} (v - b_t)$.                           ▷ Use (14) for $\text{prox}_{\sigma_A}$
7:         $b_t = \text{prox}_{\sigma_B} (v - a_t)$.                           ▷ Use (21) for $\text{prox}_{\sigma_B}$
8:         $\beta_t = \frac{1 + \sqrt{1 + 4\beta_{t-1}^2}}{2}$
9:         $\hat{a}_t = a_t + \frac{\beta_{t-1} - 1}{\beta_k} (a_t - a_{t-1})$
10:       $\hat{b}_t = b_t + \frac{\beta_{t-1} - 1}{\beta_k} (b_t - b_{t-1})$
11:     **end for**
12:     **return** $z = v - a_S - b_S$
13: **end function**

---

## D.2 Subgradient Descent Algorithm With Approximate Projection

We now use the approximate projection on $\mathcal{X}$ computed in Algorithm 1 to solve (8). In particular we analyze the following algorithm

---

**Algorithm 2** Approximate Projected sub-gradient descent

---

1: **function** APPROX-PROJ-SUBG($\mathbf{K}_0, L, R, \hat{R}, T$)
2:     $s = \frac{L}{\sqrt{T}} \cdot (\sqrt{R^2 + \hat{R}^2}$                               ▷ compute stepsize
3:     Initialize $t_0 = 1$.
4:     **for** $t = 1, \ldots, T$ **do**
5:         compute $h_{t-1}$                               ▷ subgradient of $f$ at $\mathbf{K}_{t-1}$
6:         $v_t = \mathbf{K}_{t-1} - \frac{s}{\|h_{t-1}\|} h_{t-1}$
7:         $\tilde{\mathbf{K}}_t = IIP_F ISTA(v_t, \sqrt{T})$                       ▷ Use Algorithm 1
8:         $\mathbf{K}_t = Proj_A(\tilde{\mathbf{K}}_t)) = \mathbf{K}_t - \text{prox}_{\sigma_A}(\mathbf{K}_t)$              ▷ Use (14)
9:     **end for**
10: **end function**

---

## D.3 Support Functions and their Proximal operators

### D.3.1 Support functions

The expressions for the support functions $\sigma_A$ and $\sigma_B$ are provided below.

**Claim 1.** $\sigma_A(a) = n \ \max(\lambda_{\max}(a), 0)$

*Proof.* $\sigma_A(a) = \max_{\mathbf{K} \in A} \ \text{tr}(a^\top \mathbf{K})$. The Eigen decomposition of $a$ gives $a = \sum_i \lambda_i \mathbf{u}_i \mathbf{u}_i^\top = \mathbf{U}\Lambda\mathbf{U}^\top$, where the $\mathbf{u}_i$'s are chosen to be forming an orthogonal basis. The matrix $\mathbf{K}$ can be written

using this basis as $\mathbf{U}\mathbf{M}\mathbf{U}^\top$, where $\mathbf{M}$ need not be a diagonal one.

$$\mathrm{tr}(a^\top\mathbf{K}) = \mathrm{tr}(\mathbf{U}\mathbf{M}\mathbf{U}^\top\mathbf{U}\Lambda\mathbf{U}^\top) = \mathrm{tr}(\mathbf{U}\mathbf{M}\Lambda\mathbf{U}^\top) = \mathrm{tr}(\mathbf{M}\Lambda) \leq \sum_{i:\lambda_i>0} M_{ii}\lambda_i, \qquad (13)$$

since $M_{ii}$ need to be non-negative. Now,

$$\max_{\mathbf{K}\in\mathcal{S}} \mathrm{tr}(a^\top\mathbf{K}) \leq \max_{M_{ii}>0,\ \sum_i M_{ii}\leq n} \sum_{i:\lambda_i>0} M_{ii}\lambda_i = n\max(\lambda_{\max}(\mathbf{X}),0)$$

Hence, by choosing $M_{ii} = n$ for $i$ corresponding to the largest positive eigen value, or $M_{ii} = 0$, if $\lambda_i < 0, \forall i$, we get $\sigma_A(a) = n\max(\lambda_{\max}(a),0)$. $\qquad\square$

**Claim 2.** $\sigma_B(b) = \max_{K\in B}\ tr(b^\top K) = \begin{cases} tr(b) & b_{ij} = 0, \forall(i,j)\in E \\ \infty & \textit{otherwise.} \end{cases}$

*Proof.* Trivial by using the definition of the support function. $\qquad\square$

### D.3.2   Proximal operators

**Claim 3.** $prox^\alpha_{\sigma_A}(\hat{a}) = \mathbf{U}Diag(prox^\alpha_m(z))\mathbf{U}^\top$, *where* $\mathbf{U}Diag(z)\mathbf{U}^\top$ *is the eigen decomposition of* $\hat{a}$ *and* $m(z) = \max([z;0])$. *And* $prox^\alpha_m(z)|_i = \min(z_i, \max(t^*_{\max},0))$ *where* $t^*_{\max}$ *is the solution of*

$$\sum_{i=1}^n \frac{1}{\alpha}(z_i - t)_+ = 1 \qquad (14)$$

*Proof.*

$$prox^\alpha_m(z) = \operatorname*{argmin}_x \frac{1}{2\alpha}\|x - z\|_2^2 + max([x;0])$$

$$= \operatorname*{argmin}_{z;\ t\geq x_i, t\geq 0} \frac{1}{2\alpha}\|x - z\|_2^2 + t \qquad (15)$$

Let $L(x,t,\mu) = \frac{1}{2\alpha}\|x - z\|_2^2 + t - \xi t + \sum_{i=1}^n \mu_i(x_i - t)$. Equating the gradient of the Lagrangian function to 0 at optimality,

$$\frac{\partial L}{\partial x_i} = \frac{1}{\alpha}(x_i^* - z_i^*) + \mu_i^* = 0. \qquad (16)$$

$$\frac{\partial L}{\partial t} = \sum_{i=1}^n \mu_i^* + \xi^* = 1. \qquad (17)$$

The KKT optimality conditions provide

$$\xi^* t^* = 0, \mu_i^*(x_i^* - t^*) = 0 \qquad (18)$$

$\mu_i^* > 0 \Rightarrow x_i = t^*$. Combining this with the constraint $\mu_i \geq 0$ and (16) gives $\mu_i = \frac{1}{\alpha}(z_i - t^*)_+$

$$\sum_{i=1}^n \frac{1}{\alpha}(z_i - t^*)_+ + \xi^* = 1 \qquad (19)$$

$t^* > 0$ solves the above equation if and only if

$$\sum_{i=1}^n \frac{1}{\alpha}(z_i - t)_+ = 1 \qquad (20)$$

since $\xi^* = 0$ in that case. Hence $t^* = \max(t^*_{\max},0)$, where $t^*_{\max}$ is the solution for (20). And we can recover $x$ from the previous equations. $\qquad\square$

**Claim 4.** $b^* = prox^\alpha_{\sigma_B}(\hat{b}) = \operatorname{argmin}_{b\in B} \frac{1}{2\alpha}\|b - \hat{b}\|^2 + \sigma_B(b)$

$$\Rightarrow b^*_{i,j} = \begin{cases} 0 & (i,j)\in E \\ \hat{b}_{i,j} & i\neq j, (i,j)\notin E \\ \hat{b}_{i,j} - \sigma & i = j \end{cases} \qquad (21)$$

### D.4 Proof of Theorems

We define

$$\partial_\epsilon p_{\sigma_\mathcal{X}} = \left\{ z \left| \frac{1}{2}\|z - v\|^2 + x^\top z \le \min_{z \in \mathbb{R}^n} p_{\sigma_\mathcal{X}}(z; v) + \epsilon, \quad \forall x \in \mathcal{X} \right. \right\} \tag{22}$$

we make the following claims.

**Claim 5.** *If $z = v - P_\mathcal{X}^\epsilon(v) \in \partial_\epsilon p_{\sigma_\mathcal{X}}(v)$, defined in (22), then*

$$\|P_A^\epsilon(v) - P_A(v)\| \le \sqrt{2\epsilon} \tag{23}$$

*For any $x \in \mathcal{X}$*

$$\|P_A^\epsilon(v) - x\|^2 \le \|v - x\|^2 + \epsilon \tag{24}$$

*Proof.* To prove (23), note that $p_{\sigma_\mathcal{X}}$ is strongly convex in $z$, and for any $z \in \partial_\epsilon p_{\sigma_\mathcal{X}}(v)$ the following is true

$$\epsilon \ge p_{\sigma_\mathcal{X}}(z; v) - p_{\sigma_\mathcal{X}}(z^*; v) \ge \frac{1}{2}\|z - z^*\|^2$$

where $z^* = \mathrm{prox}_{\sigma_\mathcal{X}}(v) = v - P_\mathcal{X}(v)$. Plugging $z = v - P_\mathcal{X}^\epsilon(v)$ in the above relation proves (23). To prove (24) we note that for any $z \in \partial_\epsilon p_{\sigma_\mathcal{X}}(v)$ and $x \in \mathcal{X}$

$$\frac{1}{2}\|z - v\|^2 + x^\top z \le \min_{z \in \mathbb{R}^n} p_{\sigma_\mathcal{X}}(z; v) + \epsilon \le p_{\sigma_\mathcal{X}}(0; v) + \epsilon = \frac{1}{2}\|v\|^2 + \epsilon$$

Setting $z = v - P_\mathcal{X}^\epsilon(v)$, and rearranging terms proves (24). □

**Claim 6.**

$$\mathrm{Prox}_{\sigma_\mathcal{X}}(v) = \operatorname*{argmin}_{(a,b)} p_{\sigma_\mathcal{X}}(a, b; v) \left( = \frac{1}{2}\|(a + b) - v\|^2 + \sigma_A(a) + \sigma_B(b) \right) \tag{25}$$

*Proof.* Check that $\iota_\mathcal{X}(x) = \iota_A(x) + \iota_B(x)$ and the $\sigma_\mathcal{X}(z) = \max_x x^\top z - \iota_\mathcal{X}(z)$. Following the definition of indicator function of $\sigma_\mathcal{X}$, we have

$$\min_z \frac{1}{2}\|z - v\|^2 + \sigma_\mathcal{X}(z) = \min_z \frac{1}{2}\|z - v\|^2 + \max_x \left\{ x^\top z - \iota_A(x) - \iota_B(x) \right\}$$

Introducing the support functions $\sigma_A$ and $\sigma_B$

$$= \min_z \frac{1}{2}\|z - v\|^2 + \max_x \left\{ x^\top z - \max_a \left( x^\top a - \sigma_A(a) \right) - \max_b \left( x^\top b - \sigma_B(b) \right) \right\}$$

The maximization over $a, b$ can be posed as a minimization because of the negative sign. Using strong duality, we get

$$= \min_z \frac{1}{2}\|z - v\|^2 + \max_x \min_{a,b} \left\{ x^\top (z - a - b) + \sigma_A(a) + \sigma_B(b) \right\}$$

To be dual feasible the coefficient of $x$ must be zero, leading to $z = a + b$, which is used to eliminate $z$ and we prove the claim. □

### D.5 Proof of Theorem 5 (paper)

*Proof.* Starting from $x_0 \in \mathbb{R}^{n \times n}$, let $x_k, k \ge 1$ be defined in (12). Let $y_k = x_k - \alpha_k h_k$, $r_k = \|x_k - x^*\|_F$. Then, it follows that

$$\begin{aligned}
r_{k+1}^2 = \|P_X^\epsilon(y_k) - x^*\|_F^2 &\le \|y_k - x^*\|_F^2 + \epsilon \\
&= \|x_k - \alpha_k h_k - x^*\|^2 + \epsilon \\
&= \epsilon + r_k^2 - 2\alpha_k h_k^\top (x_k - x^*) + \alpha_k^2 \|h_k\|^2 \\
&\le \epsilon + r_k^2 - 2\alpha_k (f(x_k) - f^*) + \alpha_k^2 \|h_k\|^2
\end{aligned} \tag{26}$$

The first inequality is a consequence of (24), and the last inequality is true because of convexity of $f$. If we define $f_T^* = \min\{f(x_i) \mid i \in \{0, \dots, T\}\}$, then

$$f_T^* - f^* \leq \frac{1}{2\sum_{k=0}^{T-1}\alpha_k}\left(r_0^2 + \sum_{k=0}^{T-1}\left(\epsilon + \alpha_k^2\|h_k\|^2\right)\right)$$

Under the choice of $\alpha_k = \frac{s}{\|h_k\|}$ we have

$$f_T^* - f^* \leq \frac{L}{2sT}\left(R^2 + T(\epsilon + s^2)\right)$$

Minimizing RHS as a function of $s$ yields $s = \sqrt{\frac{R^2}{T} + \epsilon}$ and thus proves Theorem 5. $\qquad\square$

### D.6 Proof of Theorem 6 (paper)

*Proof.* Check that for every $t = 1, \dots, T$, Algorithm 1 computes $\hat{a}_t, \hat{b}_t$ such that

$$p_{\sigma_{\mathcal{X}}}(\hat{a}_t, \hat{b}_t; v_t) \leq \min_{a,b \in S_n} p_{\sigma_{\mathcal{X}}}(a, b; v_t) + \frac{\hat{R}^2}{T}$$

and $\mathbf{K}_t = Proj_A(v_t - \hat{a}_t - \hat{b}_t)$. Using $\mathbf{K}^* \in A$ and the non-expansiveness of the projection operator, the following holds
$$\|\mathbf{K}_t - \mathbf{K}^*\|_F^2 \leq \|v_t - \hat{a}_t - \hat{b}_t - \mathbf{K}^*\|_F^2$$

The proof follows by retracing the steps in Theorem 5 with $\epsilon \geq \frac{\hat{R}^2}{T}$. $\qquad\square$

## E   Computation of Constants

We need to evaluate the constants involved in proposed projection method, which are necessary to compute the step size.

### E.1   Computation of $\hat{R}$

Recall that
$$(a^*(v), b^*(v)) = \operatorname*{argmin}_{a,b} \frac{1}{2}\|a + b - v\|^2 + \sigma_A(a) + \sigma_B(b) \tag{27}$$

We define
$$\hat{R}^2 \geq \|a^*(v)\|^2 + \|b^*(v)\|^2 \ \forall v = \mathbf{K} + h, \mathbf{K} \in A, h \in \partial f(\mathbf{K})$$

where
$$A = \{K \succcurlyeq 0, tr(K) = n\} \text{ and } B = \{diag(K) = 1, K_{ij} = 0 \text{ for } (i,j) \notin E\} \tag{28}$$

We have $\sigma_A(M) = \lambda_{\max}(M)$ and $\sigma_B(N) = tr(M)$ if $M_{ij} = 0$ as soon as $(i,j) \in E$, and infinity otherwise. Given a matrix $Z$, and its projection $\hat{Z}$ on $A \cap B$, we have $Y = v - \hat{v} = M + N$ with $\sigma_A(M) + \sigma_B(N)$ minimal.

We simply need to exhibit a single pair of optimizers $M, N$. For this, we may use $\tilde{A} = \{K \succcurlyeq 0\}$, for which $\sigma_{\tilde{A}}(M) = 0$ if $M \preccurlyeq 0$, and infinity otherwise.

We have $Y = M + N$ and thus $Y \preccurlyeq N$. Moreover, $tr(N) \leq n\lambda_{\max}(Y)$ because the decomposition $Y = 0 + \lambda_{\max}(Y)I$ is feasible.

Thus, $\lambda_{\min}(Y) \leq \lambda_{\min}(N) \leq \lambda_{\max}(N) \leq n\lambda_{\max}(Y)$.

This allows to show that
$$\|N\|_F^2 \leq n\|N\|_{op}^2 \leq n^3\|Y\|_{op}^2 \tag{29}$$

Now to bound the operator norm of $Y$ we proceed as follows: by choice $v = \mathbf{K} + h$ where $\mathbf{K} \in A$.

**Bound on** $\|v - \hat{v}\|_F \leq n + L$

**Proof:**

$$\|Z - \hat{Z}\| \leq \|\mathbf{K} - Proj_{A \cap B}(\mathbf{K})\| + \|h\|$$

We bound the first term as follows. By definition of projection and since $\mathbf{I}$ is feasible for both $A$ and $B$, we have

$$\|Y\| \leq \|\mathbf{K} - Proj_{A \cap B}(\hat{Z})\| \leq \|\mathbf{K} - \mathbf{I}\|$$

Squaring both sides

$$\|\mathbf{K} - Proj_{A \cap B}(\hat{Z})\|_F^2 \leq \|\mathbf{K} - \mathbf{I}\|^2 = \|\mathbf{K}\|^2 - 2\,tr(\mathbf{K}) + tr(\mathbf{I}) \leq n$$

Substituting this estimate in $\hat{R}^2 = n^3 \|Y\|_{op}^2 \leq n^5$

## E.2 Computation of R

For the sets $A$ defined in (28), note that $R = \max_{\mathbf{K} \in A} \|\mathbf{K}\|_F$. We derive a bound on the objective function as follows

$$\|\mathbf{K}\|_F^2 = tr(\mathbf{K}^2) = \sum_{i=1}^{n} \lambda_i^2(\mathbf{K}) \leq \left( \sum_{i=1}^{n} \lambda_i(\mathbf{K}) \right)^2 = \left( Tr(\mathbf{K}) \right)^2 = n^2$$

and thus the result follows.

## E.3 Computation of L

For SPORE, the subgradient is given by $-\frac{1}{2}\mathbf{Y}\alpha\alpha^\top\mathbf{Y} + \beta\mathbf{v}\mathbf{v}^\top$, which implies that

$$L \leq \max_{0 \leq \alpha_i \leq C,\, \|v\|_2 = 1} \left\| \frac{1}{2}\mathbf{Y}\alpha\alpha^\top\mathbf{Y} + \beta\mathbf{v}\mathbf{v}^\top \right\|_F$$

Using the equality $\|\mathbf{M}\|_F^2 = \sum_i \lambda_i^2(\mathbf{M})$ from Section E.2 above, and convexity, we get

$$L \leq \max_{0 \leq \alpha_i \leq C,\, \|v\|_2 = 1} \frac{1}{2}\|\alpha\|_2^2 + \beta\|\mathbf{v}\|_2^2$$

The last inequality follows from the fact that for rank 1 matrices $\mathbf{u}\mathbf{u}^\top$, $\lambda_{max}\|\mathbf{u}\|_2^2$. For the chosen constant $\beta = \frac{\vartheta(G)}{\vartheta(G)}$ from Theorem 4, note that $\beta \leq 1$, since whenever $\vartheta(G) \geq \sqrt{n}$, we can work on the complement graph and $\vartheta(G)\vartheta(\bar{G}) = n$ [6]. Thus, from the constraints on $\alpha$, it follows that $L \leq C^2\sqrt{n}$.

## F  Multiple Graph Transduction

Recalling the notations in the paper – let $\mathbb{G} = \{G^{(1)}, \ldots, G^{(M)}\}$ be a set of simple graphs $G^{(k)} = (V, E^{(k)})$, defined on a common vertex set $V = [n]$. We introduce some more notations – let

$$\mathcal{K}(\mathbb{G}) = \left\{ \mathbb{K} \mid \mathbb{K} = \{\mathbf{K}^{(1)}, \ldots, \mathbf{K}^{(M)}\}, \mathbf{K}^{(k)} \in \mathcal{K}(G^{(k)}) \right\} \tag{30}$$

and let $\tilde{\mathbf{K}}^\eta = \sum_{k \in [M]} \eta_k \mathbf{K}^{(k)}$, for $\eta \in \mathcal{S}^{M-1}$.

At this point, we would like to gather some important results before we prove the improved graph-dependent generalization bound. We define an analog of $\vartheta$ for the multiple graphs setting –

$$\vartheta(\mathbb{G}) = \min_{\mathbb{K} \in \mathcal{K}(\mathbb{G})} \min_{\eta \in \mathcal{S}^{M-1}} \bar{\omega}(\tilde{\mathbf{K}}^\eta) \tag{31}$$

where $\bar{\omega}(\cdot)$ is the 1-class SVM dual formulation, defined as –

$$\bar{\omega}(\mathbf{K}) = \max_{\alpha \in \mathbb{R}_+^n} 2\alpha^\top \mathbf{1} - \alpha^\top \mathbf{K}\alpha \tag{32}$$

We also define $k^*$, for convenience of the subsequent proofs –

$$k^* = \underset{k \in [M]}{\operatorname{argmin}} \vartheta(G^{(k)}) = \underset{k \in [M]}{\operatorname{argmax}} \vartheta(\bar{G}^{(k)}) \tag{33}$$

where $\bar{G}^{(k)}$ is the complement graph of $G^{(k)}$. The second equivalence follows from the fact that $\vartheta(G)\vartheta(\bar{G}) = n$ [6] for any graph $G$. We state some important lemmas, used in the generalization analysis. We begin with a trivial bound on the spectral norm of convex combination of graph orthonormal embeddings –

**Lemma 4.** *For any* $\mathbb{K} \in \mathcal{K}(\mathbb{G})$ *and* $\eta \in \mathcal{S}^{M-1}$,

$$\lambda_1(\tilde{\mathbf{K}}^\eta) \leq \lambda_1(\mathbf{K}^{(l^*)}) \qquad where \qquad l^* = \underset{l \in [M]}{\operatorname{argmax}} \lambda_1(\mathbf{K}^{(l)}) \tag{34}$$

*Proof.* We use convexity of the spectral norm to prove our result –

$$\lambda_1(\tilde{\mathbf{K}}^\eta) = \lambda_1\Big( \sum_{k \in [M]} \eta_k \mathbf{K}^{(k)} \Big) \leq \sum_{k \in [M]} \eta_k \lambda_1(\mathbf{K}^{(k)}) \leq \max_{\tau \in \mathcal{S}^{M-1}} \sum_{k \in [M]} \tau_k \lambda_1(\mathbf{K}^{(k)}) = \lambda(\mathbf{K}^{(l^*)})$$

where $l^*$ as in the statement of the Lemma. $\qquad \square$

We bound the spectral norm over all possible orthonormal embeddings of multiple graphs by $\vartheta$ –

**Lemma 5.** *Given a set of simple graphs* $\mathbb{G}$

$$\max_{\mathbb{K} \in \mathcal{K}(\mathbb{G})} \max_{k \in [M]} \lambda_1(\mathbf{K}^{(k)}) = \vartheta(\bar{G}^{(k^*)})$$

*where* $\bar{G}^k$ *is the complement graph of* $G^k$, *and* $k^*$ *as in* (33).

*Proof.* From Lemma 2 (paper), it follows that $\max_{\mathbf{K} \in \mathcal{K}(G)} \lambda_1(\mathbf{K}) = \vartheta(\bar{G})$. Thus, we get

$$\max_{\mathbf{K}^{(k)} \in \mathcal{K}(G^{(k)}),\, k \in [M]} \max_{k \in [M]} \lambda_1(\mathbf{K}^{(k)}) = \max_{k \in [M]} \max_{\mathbf{K} \in \mathcal{K}(G^{(k)})} \lambda_1(\mathbf{K}) = \max_{k \in [M]} \vartheta(\bar{G}^{(k)})$$

The proof follows from the definition of $k^*$ in (33). $\qquad \square$

Similar to Lemma 3 (paper), we relate MKL formulation to $\vartheta$ –

**Lemma 6.** *Given a set of simple graphs* $\mathbb{G} = \{G^{(1)}, \ldots, G^{(M)}\}$ *defined on a common vertex set* $V$, *let* $\mathbf{y} \in \mathcal{Y}^n$ *be the unknown binary labels. Then, for any subgraph* $S \subseteq V$,

$$\min_{\mathbb{K} \in \mathcal{K}(\mathbb{G})} \min_{\eta \in \mathcal{S}^{M-1}} \omega_C(\tilde{\mathbf{K}}^\eta, \mathbf{y}_S) \leq \vartheta(\mathbb{G})/2$$

*where* $\vartheta(\mathbb{G})$ *as in* (31).

*Proof.* For any simple graph $G = (V, E)$, with unknown binary labels $\mathbf{y} \in \mathcal{Y}^n$ over the set $V$; we note an interesting property of orthonormal embedding that if $\mathbf{U} \in Lab(G)$, then $\tilde{\mathbf{U}} = \mathbf{UY} \in Lab(G)$, where $Y_{ij} = y_i$, for $i = j$; 0 otherwise. Also, before we proceed, we recall a trivial bound $\omega_C(\tilde{\mathbf{K}}^\eta, \mathbf{y}_S) \leq \omega_\infty(\tilde{\mathbf{K}}^\eta, \mathbf{y})$, which follows from the primal formulation of SVM. Now, we bound the quantity of interest as follows –

$$\min_{\mathbb{K} \in \mathcal{K}(\mathbb{G})} \min_{\eta \in \mathcal{S}^{M-1}} \omega_C(\tilde{\mathbf{K}}^\eta, \mathbf{y}_S) \leq \min_{\mathbb{K} \in \mathcal{K}(\mathbb{G})} \min_{\eta \in \mathcal{S}^{M-1}} \omega_\infty(\tilde{\mathbf{K}}^\eta, \mathbf{y})$$

$$= \min_{\mathbf{U}^{(k)} \in Lab(G^{(k)})} \min_{\eta \in \mathcal{S}^{M-1}} \min_{\mathbf{w}^{(k)} \in \mathbb{R}^{d_k}} \frac{1}{2} \sum_{k \in [M]} \eta_k \|\mathbf{w}^{(k)}\|_2^2 \quad \text{s.t.} \quad y_i \sum_{k \in [M]} \eta_k \mathbf{w}^{(k)\top} \mathbf{u}_i^{(k)} \geq 1,\ \forall i \in [N]$$

Note that here $\mathbf{u}_i^{(k)}$ is the $i^{th}$ column of the orthonormal embedding of $k^{th}$ graph. Also, we assume that each orthonormal embedding $\mathbf{U}^{(k)}$ is of the dimension $d_k \times n$. Now, using the property of orthonormal embedding in the beginning of the proof, we get

$$= \min_{\tilde{\mathbf{U}}^{(k)} \in Lab(G^{(k)})} \min_{\eta \in \mathcal{S}^{M-1}} \min_{\mathbf{w}^{(k)} \in \mathbb{R}^{d_k}} \frac{1}{2} \sum_{k \in [M]} \eta_k \|\mathbf{w}^{(k)}\|_2^2 \quad \text{s.t.} \quad \sum_{k \in [M]} \eta_k \mathbf{w}^{(k)\top} \tilde{\mathbf{u}}_i^{(k)} \geq 1,\ \forall i \in [N]$$

where $\tilde{\mathbf{U}}^{(k)} = \mathbf{U}^{(k)}\mathbf{Y}$. Finally, using the dual formulation of MKL proves the result. $\qquad \square$

We relate $\vartheta$ of the multiple graphs (31) to that of individual graphs –

**Lemma 7.** $\vartheta(\mathbb{G}) \le \vartheta(G^{(k^*)})$, where $\vartheta(\mathbb{G})$ as in (31), and $k^*$ as in (33).

*Proof.* Proof follows by applying an alternate definition of $\vartheta$ as in [4] –

$$\vartheta(G) = \min_{\mathbf{K} \in \mathcal{K}(G)} \bar{\omega}(\mathbf{K})$$

where $\omega(\cdot)$ as in (32). Now, expanding $\vartheta(\mathbb{G})$, we get

$$\vartheta(\mathbb{G}) = \min_{\mathbf{K}^{(k)} \in \mathcal{K}(G^{(k)}), \forall k \in [M]} \ \min_{\eta \in \mathcal{S}^{M-1}} \bar{\omega}\Big( \sum_{k \in [M]} \eta_k \mathbf{K}^{(k)} \Big)$$

$$\le \min_{\mathbf{K}^{(k)} \in \mathcal{K}(G^{(k)}), \forall k \in [M]} \ \min_{k \in [M]} \bar{\omega}\big(\mathbf{K}^{(k)}\big)$$

$$\le \min_{k \in [M]} \Big( \min_{\mathbf{K} \in \mathcal{K}(G^{(k)})} \bar{\omega}\big(\mathbf{K}\big) \Big)$$

Using the definition of $\vartheta$ above, and $k^*$ as in (33) proves the result. $\qquad\square$

Now, we prove the main result on multiple graph transduction. Let $\mathbb{K}^*$ be the optimal graph embeddings computed from (17) (paper). Let $\alpha^*, \eta^*$ be the solution to $\min_{\eta \in \mathcal{S}^{M-1}} \omega_C(\tilde{\mathbf{K}}^{*\eta}, \mathbf{y}_S)$, then the final predictions of (17) (paper) is given by $\hat{y}_i = \sum_{j \in S} \eta_k^* K^{*(k)}_{ij} \alpha_j^* y_j,\ \forall i \in [n]$. The proposed MKL style solution allows is to extend Theorem 4 (paper) to the multiple graphs setting –

**Theorem 8.** *Given a set of simple graphs $\mathbb{G}$ and labels of a randomly drawn subgraph $S \subset V$, $m = |S|$; let $\hat{\mathbf{y}}$ be the predictions learnt by MKL-SPORE (17) for parameters $C = \Big( \frac{\vartheta(\mathbb{G})}{m\sqrt{\vartheta(\bar{G}^{(k^*)})}} \Big)^{\frac{1}{2}}$ and $\beta = \frac{\vartheta(\mathbb{G})}{\vartheta(\bar{G}^{(k^*)})}$, where $k^*$ as in (33). Then, for $m \le n/2$, with probability $\ge 1 - \delta$ over the choice of $S \subseteq V$, such that $|S| = m$*

$$er_{\bar{S}}^{0\text{-}1}[\hat{\mathbf{y}}] = O\Big( \frac{1}{m} \Big( \sqrt{n\vartheta(\mathbb{G})} + \log\frac{1}{\delta} \Big) \Big)^{\frac{1}{2}}$$

*where $\vartheta(\mathbb{G})$ as in (31).*

*Proof.* Let $\mathbb{K}^* = \{\mathbf{K}^{*(1)}, \ldots, \mathbf{K}^{*(M)}\}$, $\eta^*$ be the kernels learnt by MKL-SPORE (17) (paper). Let $\tilde{\mathbf{K}}^{*\eta^*} = \sum_{k \in [M]} \eta_k^* \mathbf{K}^{*(k)}$. Applying Theorem 1 (paper) for the final predictions $\hat{\mathbf{y}}$, we obtain

$$er_{\bar{S}}^{0\text{-}1}[\hat{\mathbf{y}}] \le \frac{1}{m} \sum_{i \in S} \ell^{hng}(y_i, \hat{y}_i) + 2C\sqrt{2\lambda_1\big(\tilde{\mathbf{K}}^{*\eta^*}\big)} + O\Big( \sqrt{\frac{1}{m}\log\frac{1}{\delta}} \Big)$$

Using $\lambda_1\big(\tilde{\mathbf{K}}^{*\eta^*}\big) \le \lambda_1\big(\mathbf{K}^{*(l^*)}\big)$ from Lemma 4, where $l^*$ as in (34) and $\lambda_1\big(\mathbf{K}^{*(l^*)}\big) \le \vartheta\big(\bar{G}^{(l^*)}\big)$ from Lemma 2 (paper), we get

$$\le \frac{1}{m} \sum_{i \in S} \ell^{hng}(y_i, \hat{y}_i) + 2C\sqrt{2\vartheta\big(\bar{G}^{(l^*)}\big)} + O\Big( \sqrt{\frac{1}{m}\log\frac{1}{\delta}} \Big) \qquad (35)$$

where $\bar{G}^{(l^*)}$ is complement graph of $G^{(l^*)}$. Using a similar argument as in the proof of Theorem 4 (paper), using the primal formulation of (3) (paper), we get

$$C\sum_{i \in S} \ell^{hng}(y_i, \hat{y}_i) \le \omega_C(\tilde{\mathbf{K}}^{*\eta^*}, \mathbf{y}_S) \le \Phi_{C,\beta}(\mathbb{G}, \mathbf{y}_S)$$

$$= \min_{\mathbb{K} \in \mathcal{K}(\mathbb{G})} \Big( \min_{\eta \in \mathcal{S}^{M-1}} \omega_C(\tilde{\mathbf{K}}^\eta, \mathbf{y}_S) + \beta \max_{k \in [M]} \lambda_1(\mathbf{K}^{(k)}) \Big)$$

$$\le \min_{\mathbb{K} \in \mathcal{K}(\mathbb{G})} \min_{\eta \in \mathcal{S}^{M-1}} \omega_C(\tilde{\mathbf{K}}^\eta, \mathbf{y}_S) + \beta \max_{\mathbb{K} \in \mathcal{K}(\mathbb{G})} \max_{k \in [M]} \lambda_1(\mathbf{K}) \le \frac{\vartheta(\mathbb{G})}{2} + \beta\vartheta\big(\bar{G}^{(k^*)}\big)$$

The last inequality follows from Lemma 5 and Lemma 6. Plugging back in (35) –

$$er_S^{0\text{-}1}[\hat{\mathbf{y}}] \leq \frac{1}{2Cm}\vartheta(\mathbb{G}) + \frac{\beta}{Cm}\vartheta(\bar{G}^{(k^*)}) + 2C\sqrt{2\vartheta(\bar{G}^{(k^*)})} + O\left(\sqrt{\frac{1}{m}\log\frac{1}{\delta}}\right)$$

where the last inequality follows by using $\vartheta(\bar{G}^{(l^*)}) \leq \vartheta(\bar{G}^{(k^*)})$, from the definition of $k^*$ (33). Choosing $\beta$ such that $\frac{\beta}{Cm}\vartheta\left(\bar{G}^{(k^*)}\right) = 2C\sqrt{2\vartheta\left(\bar{G}^{(k^*)}\right)}$ and optimizing for $C$ gives us the choice of parameters as in the statement of the theorem. Plugging back in (35), we get

$$= O\left(\frac{1}{\sqrt{m}}\left(\sqrt{\vartheta(\mathbb{G})\sqrt{\vartheta(\bar{G}^{(k^*)})}} + \sqrt{\log\frac{1}{\delta}}\right)\right) = O\left(\frac{1}{m}\left(\vartheta(\mathbb{G})\sqrt{\vartheta(\bar{G}^{(k^*)})} + \log\frac{1}{\delta}\right)\right)^{\frac{1}{2}}$$

where the last inequality follows from concavity $\sqrt{a} + \sqrt{b} \leq \sqrt{2(a+b)}$. Finally, using $\vartheta(\mathbb{G}) \leq \vartheta(G^{(k^*)})$ from the definition of $k^*$, and $\vartheta(G^{(k^*)})\vartheta(\bar{G}^{(k^*)}) = n$ [6] proves the result. $\qquad\square$

We can also apply the proposed algorithm in Section 4 to solve for (17) efficiently. Let $\mathbf{Y}$ be a diagonal matrix such that $Y_{ii} = y_i$, for $i \in S$, and 0 otherwise. The subgradient at $t^{th}$ iteration, for the $k^{th}$ graph is given by $\partial_{\mathbf{K}_t^{(k)}}\bar{g}(\mathbb{K}_t) = -\frac{1}{2}\eta_{tk}\mathbf{Y}\alpha_t\alpha_t^\top\mathbf{Y} + \mathbb{1}[k = l^*]\beta\mathbf{v}_t\mathbf{v}_t^\top$, where $\eta_t$, $\alpha_t$ are the solutions returned by MKL $\min_{\eta\in\mathcal{S}^{M-1}}\omega_C(\tilde{\mathbf{K}}_t^{\eta}, \mathbf{y}_S)$, and $\mathbf{v}_t = \mathrm{argmax}_{\mathbf{v}\in\mathbb{R}^n, \|\mathbf{v}\|=1}\mathbf{v}^\top\mathbf{K}_t^{(l^*)}\mathbf{v}$ is maximum Eigen vector of $\mathbf{K}_t^{(l^*)}$, where $l^* = \mathrm{argmax}_{l\in[M]}\lambda_1(\mathbf{K}_t^{(l)})$ as in Lemma 4.

## Footnotes

[1] $(a)_+ = \max(a, 0)$.