[Reviews · NeurIPS 2015]

Submitted by Assigned_Reviewer_1

The paper starts by introducing the background and showing the contributions. The authors then use the Zhang and Ando result showing that SSL reduces to an equivalent kernel-based supervised learning problem for the rest of the paper. In section 2 and appendix, they provide an error bound showing that minimising the spectral norm of the graph kernel matrix is a good way to improve generalisation. In section 3, they add a spectral norm regularization term to the Zhang and Ando formulation, and provide links between the error convergence and the Lovasz number of the data graph. Section 4 proposes a proximal solver, where the specificity is that it deals with the projection step approximately to handle the constraint of a cone-polytope intersection. Section 5 proposes an MKL extension, and section 6 concludes with experimental results on several datasets.

Overall, the paper is interesting and contains a lot of material (maybe even a bit dense for a single paper), with several important contributions. The link between large-scale topological properties of the data graph (Lovasz number), which reflect its organisation in some space or manifold, and the convergence of the embedding estimation (eq. 5) is particularly enlightening and bridges several fields.

In section 1 after equation 3 I would perhaps make explicit that the correspondance between eqs. (3) and (1) means that a semi-supervised learning problem becomes a supervised learning problem, so that the learning is on labelled samples, otherwise it is a bit surprising to see indices over S only (with no \bar{S}) in the sums in equation (3), whereas in eq. 1 the regulariser is also on non-labeled samples while the loss is on labelled data.

In section 2, The notation of equation 5 is confusing - why is there an equal sign just before \omega_C ?

In section 3 around line 162, maybe mention how the embedding (U) is recovered from K ? There are several (practical) choices for the factorisation.

In section 4,

Algorithm 1 step 8-9 - ensuring that K_r is PSD typically quite expensive (for general dense square symmetric matrices), and here an eigendecomposition is done every time (supplementary D.3.2). Is this where the algorithm takes most time? Would it be feasible to just optimise on a factorisation of K (say, Cholesky with K=LL^T), projection to the feasible set would be to a matrix with all-zeros in upper triangular, which would by construction ensure that K is PSD ?

Zhang and Ando is in NIPS 2005, not 2006

Summary: This is an interesting paper showing how spectral regularization impacts error bounds on graph transduction using orthonormal embedding, which provides new theoretical insights and convincing experimental results.

Submitted by Assigned_Reviewer_2

The paper addresses the problem of learning labels on the vertices of a graph. It builds on the work of Ando and Zhang which suggested that embedding the graph on a unit sphere, by means of an orthonormal presentation, results in better generalization error, i.e., error on the test set. Ando and Zhang however, did not present a concrete solution on how to choose an optimal embedding. The current paper's contribution is the proposal of a new optimization formulation which jointly optimize the embedding and the learned labels by introducing a regularization term. The regularization term, the spectral norm of the corresponding graph kernel matrix, is chosen based on the insight obtained from Theorem 1 in the paper which states an upper bound on the generalization error. Further results in the paper states asymptotic upper bound on the generalization error of the proposed method which has favorable scaling compared to previous results. This upper bound can also be translated to an upper bound for labeled sample complexity. While the proposed optimization can be cast an SDP the complexity of solving SDPs are not desirable. Hence, the paper suggest optimizing this problem using inexact proximal method which can scale to graphs with 1000s of vertices. Some experimental results indicate the superiority of the proposed method compared to previous methods.

The reviewer was not able to understand the main result of the paper, i.e., Theorem 4, at the high level. In the paragraph preceding Theorem 4 the paper states "in the presence of unlabeled nodes, without any assumption on the data, it is impossible to learn labels. Following existing literature [1, 12], we assume an edge links similar instances." However, there is no definition of what \emph{similar} means and there is no mention of it in the assumptions of the Theorem. The reviewer could not find any mention or use of such assumption in the proof of the Theorem either. Despite this, the claim of the theorem can be roughly stated, in the case of random graphs, as

er^{0-1}[\hat{y}]= O( 1/m (n^{3/4} + \log{1/\delta} )^{1/2} ).

This, as far as the reviewer understands, means that if vertices of a large random graph are {0,1} labeled at random and then a diminishing fraction, e.g., \omega{n^{3/4}}, of these labels are observed the algorithm is guaranteed with high probability to find the remaining labels with small error.
Summary: The reviewer was not able to understand the main result (see below) and can't make any judgment.

Submitted by Assigned_Reviewer_3

The authors look of optimizing the orthonormal embedding of a graph and the resulting impact of learning functions on the graph.

The authors start off by reviewing material from prior work using a Laplacian and the corresponding generalization bound (eqn. 2). They then discuss their bound based on the maximum eigenvalue of the kernel matrix (eqn. 4). This bound leads to an alternate optimization criterion--equation (5); \lambda_1(K) is not differentiable, but the optimization criterion is convex (sum of convex functions).

The authors use inexact proximal methods for the solution of SEER.

In the experiments section, the authors apply SEER and MKL-SEER to multiple problems and show significant performance improvements.

Overall, this paper is very solid.

There's good theoretical justification.

Also, there are multiple intuitive leaps as the paper progresses that show cleverness in the approach--the new optimization criterion, the willingness to tackle the difficult optimization problem, and the satisfying application to real problems.

This paper is a nice advancement over the prior approaches.

Summary: The authors look at the problem of finding an optimal embedding for graph transduction.

They derive

a novel method and demonstrate it works well in practice -- this paper is complete and well-explained.

Submitted by Assigned_Reviewer_4

The paper proposes a new framework for spectral regularized embedding, applied to graph transduction. The framework itself, and in particular the benefits for having an orthonormal embedding might be described more convincingly, but this is not a serious issue.

The authors then propose an interesting and elegant algorithm to solve the SEER problem approximately. This is an original contribution and it seems to provide a good alternative to solve the complex learning problem.

Experimental results tend to demonstrate the benefits of the proposed solution. It would be great to show results that demonstrate the benefits of the SEER framework, or of the orthonormal representation defined by (3) at first, and also to show the approximation accuracy of the IIP solution to the SEER problem. This is not visible in the current results, and it would be of high interest to position the actual contribution in a more global context.

The multiple graph transduction is unfortunately described at a very high level. It is definitely an interesting extension, but more details, motivation and intuitions are necessary to understand clearly how and why the proposed method is also interesting for multiple graph, and especially to understand how that algorithm is implement in such settings. In the current form, it is difficult to grasp the main ideas in the multiple graph case, and especially to really appreciate the experimental results in this case. In that respect, the last column of Table 4 (noisy case) is not very intuitive - it seems that one graph is better than multiple ones, which would contradict the motivation for dealing with multiple graphs.

Finally, the paper is not always easy to read and pretty dense at places. It also refers at too many places to the supplementary material. It is probably a sign that the authors wants to give too much information in a single paper, which actually penalises the accessibility and probably visibility of the work.
Summary: The paper proposes a new framework, SEER for orthonormal embeddings that is applied to graph transduction. An interesting algorithm based on IIP is proposed to solve the learning problem in SEER, and relatively convincing results are proposed.

Author Feedback
Author rebuttal: Masked Reviewer ID: Assigned_Reviewer_1

Q1) The paper is interesting and contains a lot of material (maybe even a bit dense for a single paper), with several important contributions.
R1) Agreed, we plan to write an extended journal version.

Q2) In section 1, after (3) I would perhaps make explicit that the correspondence between (3) and (1) means that a semi-supervised learning problem becomes a supervised learning problem, so that the learning is on labelled samples, otherwise it is a bit surprising to see indices over S only (with no \bar{S}) in the sums in (3), whereas in eq. 1 the regularizer is also on non-labeled samples while the loss is on labelled data.
R2) Good suggestion, will make this explicit in the paper.

Q3) In section 2, the notation of equation 5 is confusing - why is there an equal sign just before \omega_C?
R3) The function g(K) is defined as \omega_C plus the spectral norm regularizer, we will rephrase and clarify.

Q4) In section 3 around line 162, maybe mention how the embedding (U) is recovered from K? There are several (practical) choices for the factorization.
R4) Yes, we will make a note in the paper.

Q5) In section 4, Algorithm 1 step 8-9 - ensuring that K_r is PSD quite expensive, and here an eigendecomposition is done every time (supp D.3.2). Is this where the algo takes most time? Could optimize on a factorization of K (say, Cholesky with K=LL^T), projection to the feasible set would be to a matrix with all-zeros in upper triangular?
R5) Yes, this is our performance bottleneck. The suggested alternative is an interesting idea, we are not sure at this point. Will explore more in this direction.

Q6) Zhang and Ando is in NIPS 2005, not 2006.
R6) Apologies, will fix in the final manuscript.

------------------
Masked Reviewer ID: Assigned_Reviewer_2

Q1) The reviewer was not able to understand Theorem 4 at the high level.
R1) The main implication of the result is that we relate the error bound to graph topological measure - theta, which then leads to connecting to structural properties of the graph, lines 204 - 206.

Q2) "in the presence of unlabeled nodes, without any assumption on the data, it is impossible to learn labels. Following existing literature [1, 12], we assume an edge links similar instances." - there is no definition of what \emph{similar} means and there is no mention of it in the assumptions of the Theorem. The reviewer could not find any mention or use of such assumption in the proof of the Theorem either.
R2) We are working with unweighted graphs, and presence of an edge would mean two nodes are similar. Optimal Orthonormal embeddings tend to embed vertices to nearby regions if they have edges in between. Hence, the notion of similarity is implicit in the embedding. Will make a note in the paper.

------------------
Masked Reviewer ID: Assigned_Reviewer_4

Q1) Experimental results - it would be great to show results that demonstrate the benefits of the SEER framework, or of the orthonormal representation defined by (3) at first, and also to show the approximation accuracy of the IIP solution to the SEER problem.
R1) Preliminary experiments have shown that the SDP solver takes a lot of time for SEER to converge. However, we have shown a detailed comparison for Lovasz-theta in Table 1 (supplementary material), illustrating the convergence rate and accuracy of the proposed solution. Will include more details in the final draft.

Q2) Multiple graph transduction - more details, motivation and intuitions are necessary to understand clearly how and why the proposed method is also interesting for multiple graph, and to understand how that algorithm is implemented. Noisy case, Table 4 is not very intuitive - it seems that one graph is better than multiple ones, which would contradict the motivation for dealing with multiple graphs.
R2) Agreed, we have moved some of the details to the supplementary material, Section G due to space constraint. Will incorporate the suggested modifications in the paper.
The aim of the noisy case experiment is to show that MKL can extract the correct graph, and hence the labels even in the presence of noisy graphs; and in the absence of noise (Columns 1 and 2, Table 4), MKL strictly improves.

Q3) The paper is not always easy to read and pretty dense at places, refers at too many places to the supplementary material. Too much information in a single paper, which actually deserves accessibility and visibility of the work.
R3) Agreed, we plan to write an extended journal version.

------------------
Masked Reviewer ID: Assigned_Reviewer_5

Q1) A minus is lack of comparison of computation time in the experiments.
R1) Table 1 (supplementary material) shows a detailed comparison in time and accuracy between an SDP solver and the proposed IIP method for Lovasz-theta. Will include more details in the final manuscript.